# Interleukin-17, C-reactive protein, Neutrophil-to-Lymphocyte ratio, Lymphocyte-to-Monocyte ratio, and lipid profiles in healthy menopausal women with or without hot flashes: A cross-sectional study

**Nazila Didevar**[1☺], **Parvaneh Rezasoltani**[1☺*], **Arash Pourgholaminejad**[2☺], **Ehsan Kazemnezhad Leyli**[3☺], **Tahereh Seyednoori**[1‡], **Ziba Zahiri Sorouri**[4‡]

**1** Department of Midwifery, School of Nursing and Midwifery, Guilan University of Medical Sciences, Rasht, Iran, **2** Department of Immunology, School of Medicine, Guilan University of Medical Sciences, Rasht, Iran, **3** Department of Biostatistics, School of Health, Guilan University of Medical Sciences, Rasht, Iran, **4** Department of Obstetrics and Gynecology, Reproductive Health Research Center, Alzahra Hospital, School of Medicine, Guilan University of Medical Sciences, Rasht, Iran

☺ These authors contributed equally to this work.
‡ These authors also contributed equally to this work
* rezasoltani@gums.ac.ir, rezasoltani49@gmail.com

## Abstract

### Introduction

The reciprocation between systemic inflammatory markers (SIMs), dyslipidemia, and hot flashes (HFs) can play a part in the pathogenesis of endothelial dysfunction through menopause. This study intended to determine the association between some SIMs, lipids, and HFs in healthy menopausal women.

### Materials and methods

We designed a cross-sectional study in which 160 healthy menopausal women aged 40–60 were enrolled. Concerning their HFs status, they were stratified into two groups by consecutive sampling: without HFs (n = 40) and with HFs (n = 120). In addition to clinical variables and HFs experience, we measured the fasting serum levels of SIMs and lipid profiles (LPs), including Interleukin-17 (IL-17), high- sensitivity C-Reactive Protein (hs-CRP), Total Cholesterol (TC), Triglycerides (TG), Low-Density Lipoprotein Cholesterol (LDL-C), and High-Density Lipoprotein Cholesterol (HDL-C) in each group. Then, we calculated TC/HDL-C concerning the related variables and determined Neutrophil-to-Lymphocyte Ratio (NLR), and Lymphocyte-to-Monocyte Ratio (LMR), according to Complete Blood Count (CBC) quantitative parameters in each group. Furthermore, we used logistic regression analysis to assess the association between SIMs, LPs, and HFs.

**Data Availability Statement:** All relevant data are within the paper and its Supporting Information files.

**Funding:** This project was supported by the Vice Chancellor of Research and Technology of Guilan University of Medical Sciences (GUMS), Rasht, Iran, funding awarded to PR. The funders had no role in study design, data collection and analysis, decision to publish, or preparation of the manuscript.

**Competing interests:** The authors have declared that no competing interests exist.

## Settings

We performed this study in a governmental teaching hospital, Guilan/Rasht, Iran, from April to September 2021.

## Results

The two groups of menopausal women without and with HFs were not significantly different regarding the median of IL-17, hs-CRP, NLR, LMR, TG, HDL-C, and TC/HDL-C, and the mean of TC and LDL-C. Based on multiple logistic regression, TG levels appeared to be associated with the incidence of HFs (B = 0.004, $P$ = 0.040, Odds Ratio:1.004, 95% CI:1.000–1.009). NLR seemed to have an increasing impact on the HFs severity, according to ordinal logistic regression (B = 0.779, $P$ = 0.005, Odds Ratio = 2.180, 95%CI:1.270–3.744). Furthermore, hs-CRP negatively correlated with TG (r = -0.189, $P$ = 0.039) and TC/HDL-C (r = -0.268, $P$ = 0.003) in menopausal women with HFs.

## Conclusion

This study indicated an association between SIMs, lipids, and HFs. These connections may suggest HFs as links between SIMs/LPs alterations and their outcomes.

## Introduction

Menopause, the permanent cessation of menses, retrospectively refers to the passage of 12 months from the last menstrual period and occurs with a mean age of 52 years [1]. This physiologic process reflects the discharge of ovarian follicles, which causes a lack of estrogen and can be associated with signs and symptoms such as hot flashes (HFs), vaginal dryness, urogenital symptoms, sleep disorders, and depression. HFs are the most critical symptom and the main reason to refer menopausal women for care [1,2]. The prevalence of this annoying symptom in late menopausal transition and early postmenopause reaches 60–80% [3]. HFs are a sudden sensation of heat in the face and chest lasting 2–4 minutes and often accompany perspiration, irritability, and palpitations [2]. The Study of Women's health Across the Nation (SWAN 2012) showed that day and night HFs were strongly associated with decreased quality of life [4].

Menopause, through declining endogenous estrogen and causing HFs, may potentially contribute to dyslipidemia, according to some experts. This approach can be reciprocal [5]. The results of the SWAN study showed an inauspicious relationship between HFs and lipid profiles (LPs) [6]. Some other studies on menopausal women, however, did not reveal such a relationship [7]. Changes in LPs directly or indirectly (by affecting oxidative stress [OS]) can affect the inflammatory process. This event may be in terms of the increased abdominal fat accumulation in postmenopausal women and therefore increased abdominal adipose tissue. This process can increase some systemic inflammatory markers (SIMs) by reducing estrogen and, consequently, causing HFs [5].

HFs, dyslipidemia/OS, some SIMs, and their interaction can play a part in the pathogenesis of endothelial dysfunction, hypertension, diabetes, and cardiovascular disorders (CVD) [5]. These ailments enhance the cost of treatment and health care [8]. Moreover, there is an increasing tendency to investigate the association of increased SIMs and pro-inflammatory cytokines (hormone-like glycoproteins produced by the immune system cells [9]) with HFs. Several studies in this field reported contradictory results [10–13]. Tumor necrosis factor-

alpha (TNF-α), Interleukin-6 (IL-6), and Interleukin-1 (IL-1) are key pro-inflammatory cytokines [14]. Their association with menopause and its outcomes has drawn more consideration from researchers. Conversely, some of these cytokines, such as interleukin-17 (IL-17), have been studied less in this field. IL-17, produced by T helper 17 (Th17) lymphocytes, can produce IL-6 by affecting endothelial cells, which may successively result in atherosclerosis, thrombosis, and CVD [15]. Menopause appears to activate IL-17 receptors on the surface of epithelial cells by diminishing estrogen [16]. Some investigations revealed an association between elevated IL-17 and decreased estrogen in menopausal women [16,17]. Others, however, did not notice such a relationship [14]. Besides, as per searching in databases, just a single study (Huang 2017) assessed the association of IL-17 with HFs and observed no significant relationship between them [10].

Researchers also consider C-reactive protein (CRP) as an indicator of systemic inflammation. Increased CRP, particularly in women, is one of the non-lipid CVD markers [7,18]. Assessing the relationship of this marker with menopause and its outcomes revealed conflicting results. Experts agreed to do more research in this field [7,13,19]. Neutrophil-to-Lymphocyte Ratio (NLR) and Lymphocyte-to-Monocyte Ratio (LMR) are also SIMs measures that are becoming increasingly popular with researchers due to their easier accessibility and lower cost. Some studies observed differences in the composition of blood leukocytes between women before and after menopause. In these studies, women over 50 (due to lower estrogen levels) showed lower neutrophil counts and higher lymphocyte counts (and lower NLR) than younger women [20]. Besides, increasing IL-17 appears to elevate neutrophil production [21–23]. Thus, there might be a controversy among the recent findings. Moreover, it makes sense to assess the relationship of NLR with HFs in menopausal women and observe changes in neutrophil count and NLR among them. Considering the alterations in hematopoiesis with decreasing estrogen levels in menopause [20], some studies also observed an increase in LMR and lymphocyte count in perimenopausal and postmenopausal women [24].

Regarding the interaction of SIMs, LPs, HFs, and the pivotal impact of SIMs and LPs on endothelial dysfunction and its outcomes, HFs might also act as a potential risk factor for these disorders. This study aimed to determine the association between some SIMs, LPs, and HFs in healthy menopausal women. We hypothesized that there was some discrepancy between SIMs and LPs levels in the two groups of menopausal women without and with HFs.

## Materials and methods

### Participants and study design

In this cross-sectional study, we compared the serum levels of IL-17, high-sensitivity CRP (hs-CRP), NLR, LMR, and LPs in menopausal women without and with HFs. The Ethics Committee of Guilan University of Medical Sciences, Rasht, Iran, approved the review (Ethics Code: IR.GUMS.REC.1399.636). Using the results asserted by Huang et al. (2017), we determined the sample size. We used the mean (SD) of IL-17, declared in women with different HFs severity, and considering the expected clinical difference (1 unit). Therefore, 40 subjects in the group without HFs and 120 subjects in the group with HFs would be necessary, concerning a confidence of 95%, a power of 80%, and an alpha value of 0.05 [10].

Using consecutive sampling, we assigned eligible women from healthy menopausal women referred to one governmental teaching hospital, Guilan/Rasht, Iran, from April to September 2021 for health management or as a companion to women admitted to the obstetrics and gynecology wards. Inclusion criteria included age 40–60 years; having natural menopause; lapse of at least 12 months since the last menstruation, according to the Stages of Reproductive Aging Workshop + 10 (STRAW+10) criteria [25]; no history of any disease, including diabetes, hypertension,

cardiovascular, respiratory, infectious, liver, kidney, and thyroid diseases; not taking any medication, including anti-hypertensive and lipid-lowering agents, immunosuppressive drugs, anti-inflammatory drugs, anti-oxidant supplements, neuropsychiatric drugs, and hormone replacement therapy; not consuming tobacco products and alcohol; BMI of 18.5–30 kg/m2. Exclusion criteria included unwillingness to cooperate during the study for any reason.

Based on Fig 1, the first author (ND) individually interviewed a total of 1987 women in a consecutive manner, according to the inclusion criteria. As the sampling proceeded, she excluded 1816 women from the study because they did not meet the inclusion criteria. Finally, she collected 171 eligible women, out of which 11 were unwilling to continue the cooperation. Therefore, a total of 160 women were enrolled in the study. According to their response to HFs status, the participating women were divided into two groups without HFs (N = 40) and with HFs (N = 120). The women with HFs, who had experienced HFs during the three months before enrollment, were given a card and asked to record the number of HFs they experienced over a two-week period as part of a pilot study. According to the modified Kupperman index, the severity of HFs in each study sample was determined by calculating mean HFs (severity 1: <3 HFs/day, severity 2: 3–9 HFs/day, severity 3: ≥10 HFs/day) [26].

We fully informed all recruited women about the study goals and the voluntary nature of their participation. Then, all of them signed a written informed consent form. Through face-to-face interviews, the demographic and reproductive information of the participating women was recorded in the related form, the face validity of which was approved by ten faculty members specialized in midwifery and reproductive health. Then, their height (cm) and weight (kg) were measured with minimum clothing and without shoes, and the body mass index (BMI) (kg/m2) was obtained by dividing weight by height squared. Moreover, their waist circumference (WC) and hip circumference (HC) were measured with a minimum of clothing using a tape measure in centimeters. WC was obtained by measuring the rim of the abdomen at the midpoint of the border between the last rib and the iliac crest without applying pressure to the soft tissues. Also, HC was obtained by measuring the largest circumference at the level of the buttocks. WC/HC was calculated by dividing WC by HC.

## Main outcome measures

The principal outcomes were serum levels of some SIMs, including IL-17, hs-CRP, NLR, and LMR, and LPs, including Total Cholesterol (TC), Triglycerides (TG), Low-Density Lipoprotein Cholesterol (LDL-C), High-Density Lipoprotein Cholesterol (HDL-C), and TC/HDL-C.

## Laboratory measurements

The blood test administrator (1 individual) of the research laboratory in the hospital took 8 ml of venous blood in the morning (8 am-10 am) and after overnight fasting from the participants. The laboratory biochemist (1 individual) separated 6 ml of serum from a blood sample (to measure IL-17, hs-CRP, estradiol, and LPs) at a maximum of 40 minutes from blood sampling time and at room temperature and utilized the remainder of the blood to measure Complete Blood Count (CBC). Blood samples were rotated in the centrifuge (Novin Medco., Iran) at a speed of 3000 rpm for 3 minutes to isolate the serum. Then, the biochemist stored sufficient serum to evaluate IL-17 at -30˚C (without thawing until assessment) and used the remainder to gauge hs-CRP, LPs, and estradiol.

## Measurement of lipid profiles

LPs were determined with Auto analyzer BT3500 (Biotecnica instruments company, Italy) by Continuous Flow Analysis (CFA) method through colorimetry mechanism. The used

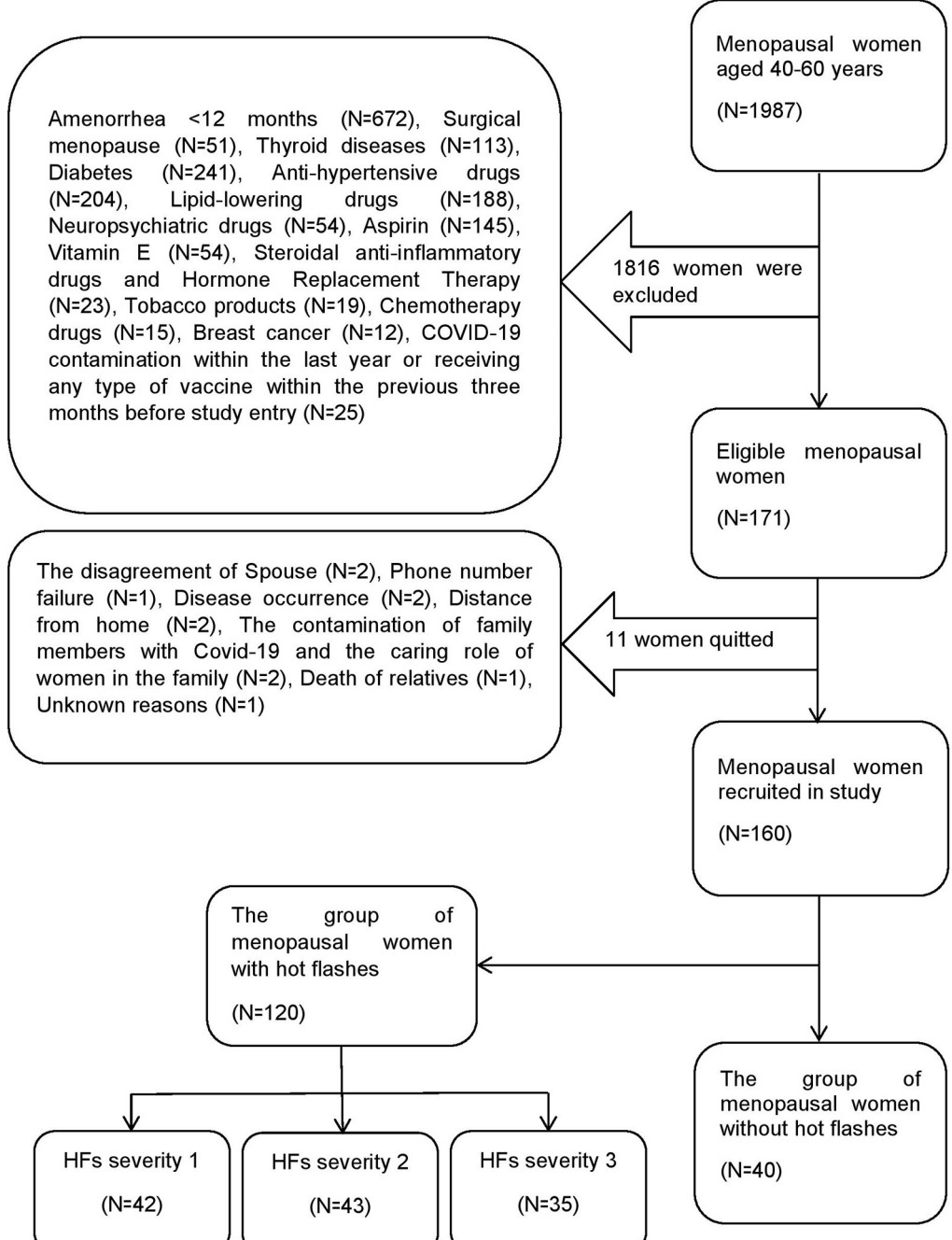

**Fig 1. Recruitment of the study samples to determine the association of SIMs and LPs with HFs.**

laboratory kits were as per the following: Cholesterol-LQ kit (CHOD-POD. Liquid, Padco company, Iran) through colorimetry for TC (mg/dl); Triglycerides kit (GPO-PAP, Pars Azmoon company, Iran) through colorimetry for TG (mg/dl); HDL-C kit (IMMUNO, Pars Azmoon company, Iran) through immune inhibition method for HDL-C (mg/dl); and Direct LDL Cholesterol kit (Pishtaz Teb company, Iran) through the direct method for LDL-C (mg/dl). TC/HDL-C is obtained by dividing TC by HDL-C. All measurements were performed according to kit manufacturers' protocols.

## Measurement of systemic inflammatory markers

IL-17 (pg/ml) was measured by the third author (AP) using ELISA reader model Bio Tek ELX 800 (Bio Tek company, USA) through spectrophotometry method and Human IL-17 ZellBio GnbH kit (ZellBio company, Germany) through Immuno Enzymo Metric Assay (IEMA) and Sandwich Ag Capture type based on the protocols of the kit manufacturer. The laboratory bio-chemist (1 individual) estimated hs-CRP (mg/l) using Auto analyzer BT3500 (Biotecnica instruments company, Italy) through Continuous Flow Analysis (CFA) and colorimetry method and Biorex Fars kit (Biorex Fars company, Iran) through immunoturbidimetric assay concerning the kit manufacturer's protocols. Furthermore, the same biochemist measured CBC parameters. Firstly, to maintain the uniformity of blood, he utilized a roller Mixer (Pole ideal Pars company, Iran) to blend the blood samples. Then, he measured the level of their quantitative parameters using the Cys Counter Sysmex KX21 (Sysmex company, Japan). Thus, he gauged neutrophil, lymphocyte, and monocyte percentages and counts. NLR was obtained by dividing neutrophil count by lymphocyte count and LMR by dividing lymphocyte count by monocyte count.

## Measurement of estradiol

Estradiol (pg/dl) was determined using ELISA microplate reader StatFax 2100 (Awareness company, USA) by spectrophotometry and Estradiol ELISA kit 96t (Ideal Tashkhis Atieh company, Iran) based on the kit manufacturer's protocols.

## Statistical analysis

The continuous variables with normal distribution were presented as the mean [standard deviation (SD)], those with non-normal distribution as the median (IQ range), and the categorical variables as frequency (percentage). Using the Shapiro-Wilk test, variables were checked for normal distribution. Regarding univariate analyses, the Chi-square test or Fisher's Exact test was used to compare the categorical baseline characteristics and the Independent-t test or Mann-Whitney U test was utilized to compare the continuous variables in menopausal women without and with HFs. Moreover, One-way analysis of variance (ANOVA) or the Kruskal-Wallis test was used to compare the continuous variables in the HFs severity-based groups. Then, Pairwise comparison and Bonferroni correction were used to determine the significant differences between HFs severity-based groups. Considering multivariate analyses, SIMs and LPs were compared in menopausal women without and with HFs using backward stepwise methods in multiple logistic regression. In addition, the ordinal logistic regression analysis was used to compare SIMs and LPs in the groups based on HFs severity. Receiver Operating Characteristic (ROC) curve and the Area Under the Curve (AUC) method, along with sensitivity and specificity, were used to determine the cut-off points for SIMs and LPs to predict the incidence and severity of HFs. SIMs' correlation with LPs in menopausal women with HFs was ascertained using Spearman's rank correlation coefficient. Data were analyzed using SPSS software version 26 (IBM Inc. USA), considering the significance level <0.05.

## Results

According to the inclusion criteria, a total of 160 menopausal women were recruited in this study. Concerning their HFs status, they were divided into one of two groups of menopausal women without HFs (N = 40) or with HFs (N = 120). Menopausal women with HFs comprised three groups of severity values of 1 (N = 42), 2 (N = 43), and 3 (N = 35) based on their HFs severity. The mean (SD) of the age of the eligible menopausal women was 54.64 (4.26) years, and the mean (SD) of the menopausal age was 48.66 (3.73) years. Table 1 presents the

**Table 1. Comparison of baseline characteristics in menopausal women without and with HFs.**

| | | Group | | P |
|---|---|---|---|---|
| | | Without HFs (n = 40) | With HFs (n = 120) | |
| Marital status N (%) | Married | 30 (75) | 91 (75.8) | 0.223[‡] |
| | Single | 0 (0) | 0 (0) | |
| | Widow | 10 (25) | 22 (18.3) | |
| | Divorced | 0 (0) | 7 (5.8) | |
| Education N (%) | Uneducated | 6 (15) | 17 (14.2) | 0.985[‡] |
| | Under diploma | 24 (60) | 71 (59.2) | |
| | Diploma | 8 (20) | 27 (22.5) | |
| | University | 2 (5) | 5 (4.2) | |
| Occupation N (%) | Housewife | 26 (65.0) | 87 (72.5) | 0.174[‡‡] |
| | Worker | 1 (2.5) | 2 (1.7) | |
| | Farmer | 2 (5.0) | 9 (7.5) | |
| | Employee (Medical sector) | 7 (17.5) | 6 (5.0) | |
| | Employee (non-medical sector) | 1 (2.5) | 9 (7.5) | |
| | Free job | 3 (7.5) | 7 (5.8) | |
| Place of residence N (%) | Urban | 33 (82.5) | 99 (82.5) | 0.999[‡] |
| | Rural | 7 (17.5) | 21 (17.5) | |
| Income status N (%) | Sufficient | 18 (45) | 65 (54.2) | 0.190[‡‡] |
| | Less than sufficient | 21 (52.5) | 55 (45.8) | |
| | More than sufficient | 1 (2.5) | 0 (0.0) | |
| Physical activities N (%) | No | 18 (45.0) | 59 (49.2) | 0.648[‡] |
| | Yes | 22 (55.0) | 61 (50.8) | |
| Type of physical activity N (%) | Walking | 20 (90.9) | 58 (95.1) | 0.558[‡] |
| | Aerobics | 2 (9.1) | 1 (1.6) | |
| | Mountain climbing | 0 (0.0) | 1 (1.6) | |
| | Yoga | 0 (0.0) | 1 (1.6) | |
| Frequency of physical activities N (%) | Twice a week | 2 (9.1) | 9 (14.8) | 0.615[‡] |
| | 3 times a week | 7 (31.8) | 23 (37.7) | |
| | Everyday | 13 (59.1) | 29 (47.5) | |
| Oily/fast foods N (%) | No | 28 (70.0) | 90 (75.0) | 0.534[‡] |
| | Yes | 12 (30.0) | 30 (25.0) | |
| Frequency of having oily & fast foods N (%) | Never | 28 (70) | 90 (75) | 0.440[‡] |
| | Once a month | 7 (17.5) | 23 (19.2) | |
| | Twice a month | 3 (7.5) | 3 (2.5) | |
| | Once a week | 2 (5) | 4 (3.3) | |
| Age (years); Mean (SD) | | 55.65 (3.63) | 54.30 (4.41) | 0.082[‡‡‡] |
| BMI (kg/m$^2$); Mean (SD) | | 26.63 (2.93) | 27.14 (2.16) | 0.319[‡‡‡] |
| WC/HC; Mean (SD) | | 0.89 (0.04) | 0.90 (0.03) | 0.583[‡‡‡] |
| Marriage duration (years); Mean (SD) | | 34.73 (8.81) | 31.52 (9.06) | 0.053[‡‡‡] |
| Physical activity duration (minutes); Mean (SD) | | 42.05 (14.53) | 50.00 (20.84) | 0.103[‡‡‡] |
| Menopausal age (years); Mean (SD) | | 49.35 (3.03) | 48.43 (3.92) | 0.216[‡‡‡] |
| Gravidity; Mean (SD) | | 4.05 (1.78) | 3.76 (1.85) | 0.267[‡‡‡‡] |
| Parity; Mean (SD) | | 3.68 (1.73) | 3.13 (1.54) | 0.067[‡‡‡‡] |
| Number of children; Mean (SD) | | 3.68 (1.73) | 3.13 (1.54) | 0.067[‡‡‡‡] |

(*Continued*)

**Table 1.** (Continued)

| | Group | | P |
|---|---|---|---|
| | Without HFs (n = 40) | With HFs (n = 120) | |
| elapsed time since the last menstruation (months); Mean (SD) | 76.20 (43.86) | 70.40 (50.74) | 0.254[####] |
| elapsed time since HFs onset (months); Mean (SD) | 00.00 (00.00) | 72.27 (50.17) | 0.261[####] |
| Estradiol (pg/ml); Median (IQR) | 8.30 (4.15–12.70) | 7.90 (4.45–16.00) | 0.757[####] |

HFs, Hot Flashes; N (%), Frequency (Percentage); SD, Standard Deviation; BMI, Body Mass Index; WC/HC, Waist Circumference/Hip Circumference; IQR, Interquartile Range

[#]Chi-squared test

[##]Fisher's exact test

[###]Independent t-test

[####]Mann-Whitney u test.

demographic and reproductive characteristics of the participants. Given the baseline characteristics, the two groups of participants without and with HFs were not significantly different. Based on Table 2, the mean age of the participants in severity values of 1, 2, and 3 were 55.85 (4.22), 54.39 (4.56), and 52.31 (3.70) years, respectively. Also, the mean elapsed time since the last menstruation and the mean elapsed time since the onset of HFs decreased from severity 1 to 3. The HFs severity-based groups were not significantly different, considering the baseline characteristics.

## Association of SIMs with HFs and their severity

According to the Mann-Whitney U test, the two groups of menopausal women without and with HFs were not significantly different in terms of IL-17, hs-CRP, NLR, and LMR levels (Table 3).

HFs severity-based groups were not significantly different in terms of the SIMs, apart from NLR, as indicated by Table 4. The median (IQR) of NLR significantly expanded in HFs severity values of 1, 2, and 3 (1.25, 1.57, 1.71 respectively; $P = 0.013$) [S1 Fig]. Given the pairwise comparison, the two groups with severity values of 1 and 2 were significantly different in terms of the median (IQR) of NLR ($P = 0.011$), according to Table 5. After Bonferroni adjustments for multiple comparisons, this difference was no more significant ($P = 0.068$). Moreover, the group with a severity value of 3 showed a higher median (IQR) of NLR compared with the group with a severity value of 1 ($P = 0.003$), which was still higher using the Bonferroni test ($P = 0.016$). After controlling age, marital duration, menopausal age, and elapsed time since the last menstruation, there was a significant association between NLR and HFs. In other words, NLR had an increasing effect on HFs severity after controlling the confounding upshots of demographic and reproductive variables ($P = 0.005$, B = 0.779, Odds Ratio = 2.180, 95%CI: 1.270–3.744). Furthermore, we observed the increasing effect of White Blood Cells (WBC) and WC/HC on HFs (Table 6).

## Association of LPs with HFs and their severity

Based on Table 3, the two groups of menopausal women without and with HFs were not significantly different regarding the levels of TC, TG, HDL-C, LDL-C, and TC/HDL-C. The mean of TC was 179.19 (35.80), 193.00 (43.71), and 201.54 (39.72) mg/dl ($P = 0.033$), and the median of TG was 133 (92.75–175), 144 (116–268), and 230 (125–349) mg/dl ($P = 0.001$), in the groups with the severity values of 1, 2 and 3 respectively (S2 and S3 Figs). Concerning the

**Table 2. Comparison of baseline characteristics in menopausal women in HFs severity-based groups.**

| | | Severity scale | | | P |
|---|---|---|---|---|---|
| | | 1 | 2 | 3 | |
| Marital status N (%) | Married | 34 (81.0) | 28 (65.1) | 29 (82.9) | 0.326ŧ |
| | Single | 0 (0) | 0 (0) | 0 (0) | |
| | Widow | 6 (14.3) | 12 (27.9) | 4 (11.4) | |
| | Divorced | 2 (4.8) | 3 (7.0) | 2 (5.7) | |
| Education N (%) | Uneducated | 2 (4.8) | 10 (23.3) | 5 (14.3) | 0.265ŧ |
| | Under diploma | 28 (66.7) | 21 (48.8) | 22 (62.9) | |
| | Diploma | 11 (26.2) | 10 (23.3) | 6 (17.1) | |
| | University | 1 (2.4) | 2 (4.7) | 2 (5.7) | |
| Occupation N (%) | Housewife | 33 (78.6) | 26 (60.5) | 28 (80.0) | 0.278ŧ |
| | Worker | 0 (0.0) | 2 (4.7) | 0 (0.0) | |
| | Farmer | 2 (4.8) | 4 (9.3) | 3 (8.6) | |
| | Employee (Medical sector) | 1 (2.4) | 2 (4.7) | 3 (8.6) | |
| | Employee (non-medical sector) | 3 (7.1) | 5 (11.6) | 1 (2.9) | |
| | Free job | 3 (7.1) | 4 (9.3) | 0 (0.0) | |
| Place of residence N (%) | Urban | 36 (85.7) | 35 (84.1) | 28 (80.0) | 0.783ŧ |
| | Rural | 6 (14.3) | 8 (18.6) | 7 (20.0) | |
| Income status N (%) | Sufficient | 23 (54.8) | 25 (58.1) | 17 (48.6) | 0.697ŧ |
| | Less than sufficient | 19 (45.2) | 18 (41.9) | 18 (51.4) | |
| | More than sufficient | 0 (0.0) | 0 (0.0) | 0 (0.0) | |
| Physical activities N (%) | No | 18 (42.9) | 23 (53.5) | 18 (51.4) | 0.588ŧ |
| | Yes | 24 (57.1) | 20 (46.5) | 17 (48.6) | |
| Type of physical activity N (%) | Walking | 23 (95.8) | 19 (95) | 16 (94.1) | 0.403ŧ |
| | Aerobics | 0 (0.0) | 0 (0.0) | 1 (5.9) | |
| | Mountain climbing | 1 (4.2) | 0 (0.0) | 0 (0.0) | |
| | Yoga | 0 (0.0) | 1 (5.0) | 0 (0.0) | |
| Frequency of physical activities N (%) | Twice a week | 5 (20.8) | 3 (15.0) | 1 (5.9) | 0.203ŧ |
| | 3 times a week | 9 (37.5) | 10 (50.0) | 4 (23.5) | |
| | Everyday | 10 (41.7) | 7 (35.0) | 12 (70.6) | |
| Oily/fast foods N (%) | No | 34 (81.0) | 33 (76.7) | 23 (65.7) | 0.290ŧ |
| | Yes | 8 (19.0) | 10 (23.3) | 12 (34.3) | |
| Frequency of having oily & fast foods N (%) | Never | 0 (0.0) | 0 (0.0) | 0 (0.0) | 0.648ŧ |
| | Once a month | 6 (75) | 8 (80.0) | 9 (75) | |
| | Twice a month | 0 (0.0) | 1 (10.0) | 2 (16.7) | |
| | Once a week | 2 (25) | 1 (10.0) | 1 (8.3) | |
| Age (years); Mean (SD) | | 55.85 (4.22) | 54.39 (4.56) | 52.31 (3.70) | 0.002ᴴ |
| BMI (kg/m$^2$); Mean (SD) | | 27.27 (2.12) | 26.56 (2.31) | 27.67 (1.90) | 0.070ᴴ |
| WC/HC; Mean (SD) | | 0.89 (0.04) | 0.89 (0.03) | 0.90 (0.02) | 0.411ᴴ |
| Marriage duration (years); Mean (SD) | | 32.11 (8.57) | 30.90 (9.45) | 31.54 (9.34) | 0.829ᴴ |
| Physical activity duration (minutes); Mean (SD) | | 47.08 (21.26) | 53.75 (20.44) | 49.70 (21.24) | 0.576ᴴ |
| Menopausal age (years); Mean (SD) | | 48.60 (4.57) | 47.95 (3.63) | 48.83 (3.43) | 0.430ᴴᴴ |

**Table 2.** (Continued)

| | Severity scale | | | P |
|---|---|---|---|---|
| | **1** | **2** | **3** | |
| Gravidity; Mean (SD) | 3.95 (2.06) | 3.53 (1.74) | 3.80 (1.75) | 0.700[ΗΗ] |
| Parity; Mean (SD) | 3.26 (1.67) | 2.98 (1.44) | 3.17 (1.52) | 0.809[ΗΗ] |
| Number of children; Mean (SD) | 3.12 (1.66) | 2.95 (1.41) | 3.11 (1.49) | 0.868[ΗΗ] |
| elapsed time since the last menstruation (months); Mean (SD) | 87.14 (54.45) | 77.30 (53.62) | 41.83 (25.27) | <0.001[ΗΗ] |
| elapsed time since HFs onset (months); Mean (SD) | 88.19 (53.70) | 78.42 (53.75) | 45.60 (25.88) | <0.001[ΗΗ] |
| Estradiol (pg/ml); Median (IQR) | 7.45 (4–12) | 8.20 (5–19) | 7.00 (4–17) | 0.853[ΗΗ] |

HFs, Hot Flashes; N (%), Frequency (Percentage); SD, Standard Deviation; BMI, Body Mass Index; WC/HC, Waist Circumference/Hip Circumference; IQR, Interquartile Range

[†]Chi-squared test

[Η]ANOVA test

[ΗΗ]Kruskal-Wallis test.

mean (SD) of TC, the groups with the severity values of 0 and 3 ($P = 0.013$) and 1 and 3 ($P = 0.017$) were significantly different, using the pairwise comparison and as indicated by Table 5. After Bonferroni adjustments for multiple comparisons, these differences were not significant anymore. Furthermore, the groups with the severity values of 1 and 2 ($P = 0.048$), 1 and 3 ($P = 0.000$), 0 and 3 ($P = 0.000$), and 2 and 3 ($P = 0.014$) were significantly different in terms of the median (IQR) of TG, using the pairwise comparison. These differences were still significant in the groups with the severity values of 1 and 3 ($P = 0.000$), and 0 and 3 ($P = 0.001$), after Bonferroni adjustments for multiple comparisons. In other words, menopausal women with a severity value of 3 had higher levels of TG [median (IQR)] than those with severity values of 1 and 0.

**Table 3. Comparison of SIMs, and LPs in menopausal women without and with HFs.**

| | Group | | P |
|---|---|---|---|
| | **Without HFs (n = 40)** | **With HFs (n = 120)** | |
| IL-17 (pg/ml); Median (IQR) | 41.21 (33.00–66.19) | 47.38 (29.91–75.35) | 0.830[Η] |
| hs-CRP (mg/l); Median (IQR) | 3.00 (2.00–5.50) | 2.75 (2.00–6.00) | 0.351[Η] |
| NLR; Median (IQR) | 1.45 (1.17–1.76) | 1.47 (1.16–1.89) | 0.610[Η] |
| LMR; Median (IQR) | 7.27 (5.79–9.13) | 8.00 (6.00–11.42) | 0.278[Η] |
| TC (mg/dl); Mean (SD) | 176.70 (34.13) | 190.66 (40.62) | 0.095[†] |
| TG (mg/dl); Median (IQR) | 134.50 (102.00–195.50) | 149.50 (111.50–238.00) | 0.072[Η] |
| HDL-C (mg/dl); Median (IQR) | 39.00 (36.00–44.50) | 42.00 (37.00–47.50) | 0.136[Η] |
| LDL-C (mg/dl); Mean (SD) | 103.03 (27.82) | 112.12 (39.68) | 0.328[†] |
| TC/HDL-C; Median (IQR) | 4.37 (3.74–5.17) | 4.28 (3.68–5.18) | 0.917[Η] |

SIMs, Systemic Inflammatory Markers; LPs, Lipid Profiles; HFs, Hot Flashes; IL-17, Interleukin-17; IQR, Interquartile Range; hs-CRP, high-sensitivity C-Reactive Protein; NLR, Neutrophil-to-Lymphocyte Ratio; LMR, Lymphocyte-to-Monocyte Ratio; TC, Total Cholesterol; SD, Standard Deviation; TG, Triglycerides; HDL-C, High-Density Lipoprotein Cholesterol; LDL-C, Low-Density Lipoprotein Cholesterol; TC/HDL-C, TC to HDL-C Ratio

[†]Independent t-test

[Η]Mann-Whitney u test.

**Table 4. Comparison of SIMs, and LPs in menopausal women in HFs severity-based groups.**

| | Severity scale | | | *P* |
|---|---|---|---|---|
| | **1** | **2** | **3** | |
| IL-17 (pg/ml); Median (IQR) | 51.88 (35.25–83.25) | 46.63 (26–76) | 45.38 (35–75) | 0.673[H] |
| hs-CRP (mg/l); Median (IQR) | 2.00 (2–3.25) | 3.00 (2–4) | 3.00 (2–4) | 0.635[H] |
| NLR; Median (IQR) | 1.25 (1–2) | 1.57 (1–2) | 1.71 (1–2) | 0.013[H] |
| LMR; Median (IQR) | 8.29 (6–12.25) | 7.00 (6–10) | 8.25 (6–10) | 0.457[H] |
| TC (mg/dl); Mean (SD) | 179.19 (35.80) | 193.00 (43.71) | 201.54 (39.72) | 0.033[t] |
| TG (mg/dl); Median (IQR) | 133.00 (92.75–175) | 144.00 (116–268) | 230.00 (125–349) | 0.001[H] |
| HDL-C (mg/dl); Median (IQR) | 43.00 (37–48.50) | 41.00 (37–47) | 40.00 (37–47) | 0.469[H] |
| LDL-C (mg/dl); Median (IQR) | 100.00 (75.75–130.25) | 112.00 (76–140) | 112.00 (90–140) | 0.506[H] |
| TC/HDL-C; Median (IQR) | 3.95 (3.95–5) | 4.34 (4–5) | 4.78 (4–6) | 0.089[H] |

SIMs, Systemic Inflammatory Markers; LPs, Lipid Profiles; HFs, Hot Flashes; IL-17, Interleukin-17; IQR, Interquartile Range; hs-CRP, high-sensitivity C-Reactive Protein; NLR, Neutrophil-to-Lymphocyte Ratio; LMR, Lymphocyte-to-Monocyte Ratio; TC, Total Cholesterol; SD, Standard Deviation; TG, Triglycerides; HDL-C, High-Density Lipoprotein Cholesterol; LDL-C, Low-Density Lipoprotein Cholesterol; TC/HDL-C, TC to HDL-C Ratio

[t] ANOVA test

[H] Kruskal-Wallis test.

**Table 5. The pairwise comparison of NLR, TC and TG in HFs severity-based groups.**

| NLR | | | | | |
|---|---|---|---|---|---|
| Sample 1-Sample 2 | Test statistic | Std. Error | Std. test statistic | Sig. | Adj. sig.[t] |
| 0–1 | 14.072 | 10.235 | 1.375 | 0.169 | 1.000 |
| 1–2 | -25.461 | 10.051 | -2.533 | 0.011 | 0.068 |
| 1–3 | -31.767 | 10.603 | -2.996 | 0.003 | 0.016 |
| 0–2 | -11.389 | 10.177 | -1.119 | 0.263 | 1.000 |
| 0–3 | -17.695 | 10.723 | -1.650 | 0.099 | 0.593 |
| 2–3 | -6.306 | 10.547 | -0.598 | 0.550 | 1.000 |
| **TC** | | | | | |
| 0–1 | -1.088 | 10.235 | -0.106 | 0.915 | 1.000 |
| 0–2 | -16.809 | 10.177 | -1.652 | 0.099 | 0.592 |
| 0–3 | -26.500 | 10.723 | -2.471 | 0.013 | 0.081 |
| 1–2 | -15.721 | 10.051 | -1.564 | 0.118 | 0.707 |
| 1–3 | -25.412 | 10.603 | -2.397 | 0.017 | 0.099 |
| 2–3 | -9.691 | 10.547 | -0.919 | 0.358 | 1.000 |
| **TG** | | | | | |
| 0–1 | 5.267 | 10.236 | 0.515 | 0.607 | 1.000 |
| 1–2 | -19.888 | 10.051 | -1.979 | 0.048 | 0.287 |
| 1–3 | -45.738 | 10.604 | -4.313 | 0.000 | 0.000 |
| 0–2 | -14.621 | 10.177 | -1.437 | 0.151 | 0.905 |
| 0–3 | -40.471 | 10.723 | -3.774 | 0.000 | 0.001 |
| 2–3 | -25.850 | 10.547 | -2.451 | 0.014 | 0.085 |

NLR, Neutrophil-to-Lymphocyte Ratio; TC, Total Cholesterol; TG, Triglycerides; HFs, Hot Flashes

[t] Bonferroni test.

**Table 6. Results of the logistic regression (backward stepwise) and the ordinal regression comparing SIMs and LPs between two groups without HFs and with HFs, and in HFs severity-based groups.**

| Variables in equation | | B | Std. Error | Sig. | Exp. (B) | 95% CI for Exp. (B) | |
|---|---|---|---|---|---|---|---|
| | | | | | | lower | upper |
| **Step 1[‡]** | Age | 0.622 | 0.699 | 0.374 | 1.862 | 0.473 | 7.330 |
| | Marriage duration | -0.039 | 0.029 | 0.182 | 0.961 | 0.908 | 1.019 |
| | Menopausal age | -0.694 | 0.699 | 0.321 | 0.500 | 0.127 | 1.966 |
| | elapsed time since the last menstruation | -0.053 | 0.058 | 0.361 | 0.948 | 0.846 | 1.063 |
| | LMR | 0.046 | 0.040 | 0.254 | 1.047 | 0.967 | 1.134 |
| | NLR | 0.556 | 0.397 | 0.162 | 1.743 | 0.801 | 3.795 |
| | TC | 0.004 | 0.006 | 0.489 | 1.004 | 0.993 | 1.015 |
| | TG | 0.004 | 0.002 | 0.066 | 1.004 | 1.000 | 1.009 |
| | HDL-C | 0.040 | 0.028 | 0.147 | 1.041 | 0.986 | 1.100 |
| | Constant | 1.673 | 3.532 | 0.636 | 5.330 | | |
| **Step 9** | Marriage duration | -0.046 | 0.024 | 0.055 | 0.955 | 0.911 | 1.001 |
| | TG | 0.004 | 0.002 | 0.040 | 1.004 | 1.000 | 1.009 |
| | Constant | 1.865 | 0.893 | 0.037 | 6.455 | | |
| **Severity[¤] scale (1, 2, 3)** | WC/HC (>0.85/≤0.85) | 1.064 | 0.4760 | 0.025 | 2.898 | 1.140 | 7.366 |
| | Menopausal age | -0.163 | 0.0449 | 0.000 | 0.849 | 0.778 | 0.927 |
| | elapsed time since the last menstruation | -0.015 | 0.0035 | 0.000 | 0.985 | 0.978 | 0.992 |
| | WBC | 0.207 | 0.1052 | 0.049 | 1.231 | 1.001 | 1.512 |
| | NLR | 0.779 | 0.2759 | 0.005 | 2.180 | 1.270 | 3.744 |
| | TG | 0.009 | 0.0016 | 0.000 | 1.009 | 1.005 | 1.012 |
| | HDL-C | 0.037 | 0.0190 | 0.048 | 1.038 | 1.000 | 1.077 |

SIMs, Systemic Inflammatory Markers; LPs, Lipid Profiles; HFs, Hot Flashes; CI, Confidence Interval; LMR, Lymphocyte-to-Monocyte Ratio; NLR, Neutrophil-to-Lymphocyte Ratio; TC, Total Cholesterol; TG, Triglycerides; HDL-C, High-Density Lipoprotein Cholesterol; WC/HC, Waist Circumference/Hip Circumference; WBC, White Blood Cells

[‡]Multiple logistic regression

[¤]ordinal logistic regression.

As stated by Table 6, after controlling age, marriage duration, menopausal age, and the elapsed time since the last menstruation, we noted a significant association between TG and HFs ($P = 0.040$, B = 0.004, Odds Ratio = 1.004, 95%CI:1.000–1.009). Using the Odds Ratio in the exponential function formula ($e^{Bx}$) equaled 1.49. Thus, the odds of developing HFs expanded by about 50% with each 100 mg/dl increase in TG level in menopausal women. Other LPs items had no significant association with HFs.

The additive effects of TG ($P < 0.001$, B = 0.009, Odds Ratio = 1.009, 95%CI:1.005–1.012) and HDL-C ($P = 0.048$, B = 0.037, Odds Ratio = 1.038, 95%CI:1.000–1.077) were observed on HFs severity by controlling the confounding effect of demographic and reproductive variables. Regarding the other components of LPs, the HFs severity-based groups did not differ significantly. Moreover, we noticed the diminishing impact of menopausal age and the elapsed time since the last menstruation on HFs severity (Table 6).

## SIMs correlation with LPs in the group of menopausal women with HFs

In the group of menopausal women with HFs, lymphocyte count positively correlated with TG ($P = 0.007$, r = 0.243), hs-CRP negatively correlated with TG ($P = 0.039$, r = -0.189), hs-CRP also negatively correlated with TC/HDL-C ($P = 0.003$, r = -0.268), using Spearman correlation analysis. Other SIMs and LPs were not significantly correlated (Table 7).

**Table 7. Results of Spearman correlation between SIMs and LPs in menopausal women with HFs.**

| | | TC | TG | HDL-C | LDL-C | TC/HDL-C |
|---|---|---|---|---|---|---|
| Lymphocyte count | Correlation coefficient | 0.069 | 0.243 | -0.097 | -0.096 | 0.116 |
| | Sig. (2-tailed) | 0.451 | 0.007 | 0.294 | 0.296 | 0.209 |
| | N | 120 | 120 | 120 | 120 | 120 |
| NLR | Correlation coefficient | -0.119 | -0.156 | -0.073 | 0.003 | -0.092 |
| | Sig. (2-tailed) | 0.195 | 0.088 | 0.428 | 0.978 | 0.319 |
| | N | 120 | 120 | 120 | 120 | 120 |
| LMR | Correlation coefficient | 0.039 | 0.048 | -0.062 | 0.034 | 0.088 |
| | Sig. (2-tailed) | 0.675 | 0.599 | 0.502 | 0.709 | 0.341 |
| | N | 120 | 120 | 120 | 120 | 120 |
| hs-CRP | Correlation coefficient | -0.119 | -0.189 | 0.164 | -0.148 | -0.268 |
| | Sig. (2-tailed) | 0.194 | 0.039 | 0.074 | 0.107 | 0.003 |
| | N | 120 | 120 | 120 | 120 | 120 |
| IL-17 | Correlation coefficient | 0.038 | 0.141 | -0.110 | -0.123 | 0.074 |
| | Sig. (2-tailed) | 0.677 | 0.125 | 0.230 | 0.181 | 0.423 |
| | N | 120 | 120 | 120 | 120 | 120 |

SIMs, Systemic Inflammatory Markers; LPs, Lipid Profiles; HFs, Hot Flashes; NLR, Neutrophil-to-Lymphocyte Ratio; LMR, Lymphocyte-to-Monocyte Ratio; hs-CRP, high-sensitivity C-Reactive Protein; IL-17, Interleukin-17; TC, Total Cholesterol; TG, Triglycerides; HDL-C, High-Density Lipoprotein Cholesterol; LDL-C, Low-Density Lipoprotein Cholesterol; TC/HDL-C, TC to HDL-C Ratio.

## Prediction levels of TG and NLR for HFs incidence

According to the ROC curve, TG and NLR predictive levels to determine HFs incidence were statistically significant ($P<0.001$, AUC = 0.694, 95%CI:0.600–0.788; $P = 0.002$, AUC = 0.676, 95%CI:0.578–0.773, respectively) in menopausal women with HFs (Table 8). Therefore, by analyzing the sensitivity and specificity of different points of TG and NLR, we noticed the best cut-off points for TG and NLR as per the following: 143 mg/dl, with 61.5% sensitivity and 60% specificity for TG, and 1.36, with 65.4% sensitivity and 60% specificity for NLR (Fig 2).

Regarding the predictive levels of TG and NLR, we noted no significant difference in determining the prognosis of the HFs severity value of 1 (Fig 3). However, these predictive levels to assess the prediction of the HFs severity value of 2 were statistically significant ($P<0.001$, AUC = 0.681, 95%CI:0.598–0.763; $P = 0.003$, AUC = 0.634, 95%CI:0.548–0.720, respectively) [Table 8]. Similarly, by examining the sensitivity and specificity of different points of TG and

**Table 8. ROC curve of TG and NLR predictive levels to determine incidence, severity 1, severity 2, and severity 3 of HFs.**

| | | Area | Std. Error | Asymptotic sig. | Asymptotic 95% CI | |
|---|---|---|---|---|---|---|
| | | | | | Lower band | Upper band |
| Without & with HFs | TG | 0.694 | 0.048 | 0.000 | 0.600 | 0.788 |
| | NLR | 0.676 | 0.050 | 0.002 | 0.578 | 0.773 |
| HFs severity 1 | TG | 0.595 | 0.049 | 0.072 | 0.499 | 0.691 |
| | NLR | 0.527 | 0.052 | 0.607 | 0.425 | 0.629 |
| HFs severity 2 | TG | 0.681 | 0.042 | 0.000 | 0.598 | 0.763 |
| | NLR | 0.634 | 0.044 | 0.003 | 0.548 | 0.720 |
| HFs severity 3 | TG | 0.733 | 0.053 | 0.000 | 0.629 | 0.836 |
| | NLR | 0.616 | 0.055 | 0.037 | 0.509 | 0.723 |

ROC, Receiver Operating Characteristic; TG, Triglycerides; NLR, Neutrophil-to-Lymphocyte Ratio; HFs, Hot Flashes; CI, Confidence Interval.

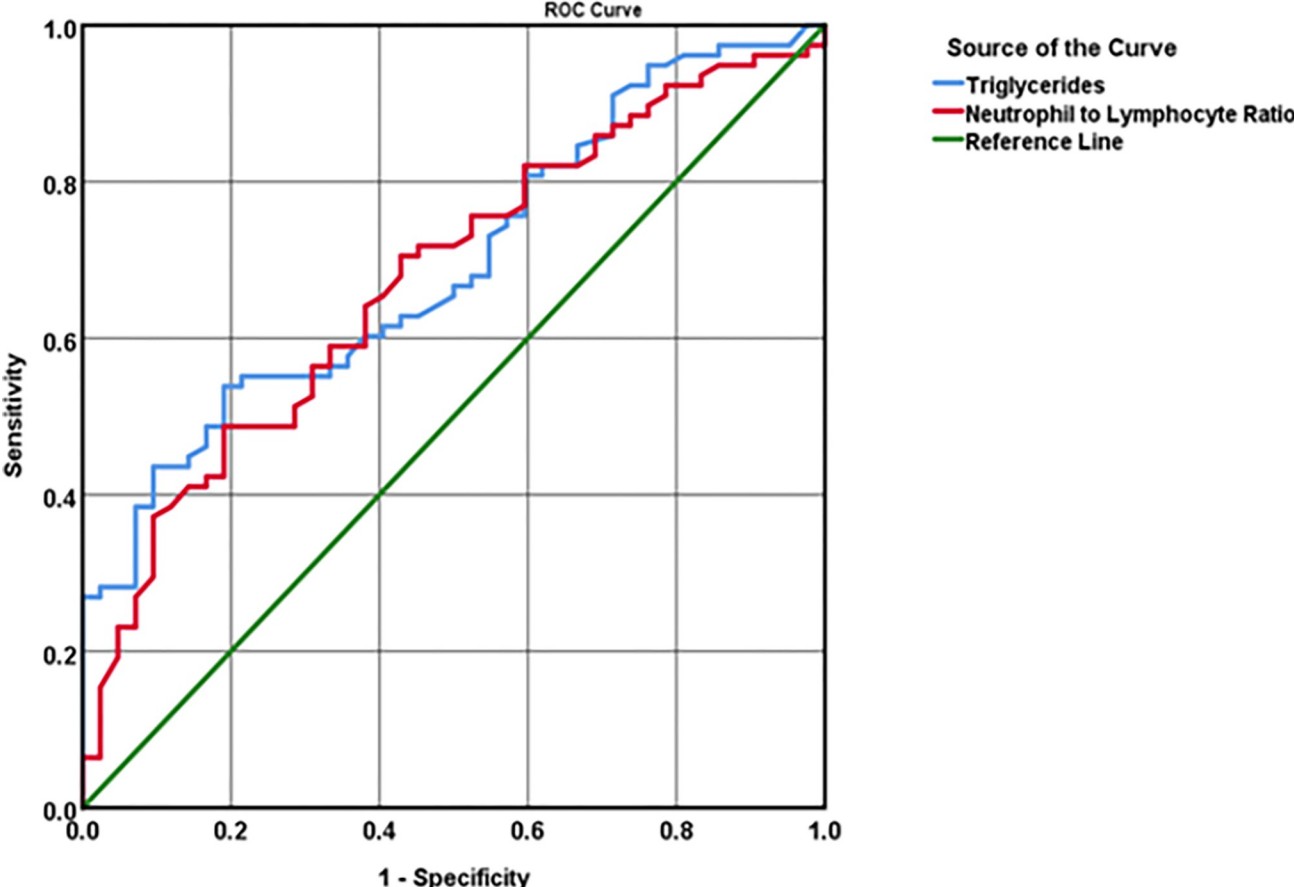

**Fig 2. ROC curve of TG (mg/dl) and NLR predictive levels to determine the incidence of HFs.**

NLR in the HFs severity value of 2, we determined the best cut-off points for TG and NLR in this group as below mentioned: 148 mg/dl with 60.3% sensitivity and 60% specificity for TG and 1.41 with 61.5% sensitivity and 56% specificity for NLR (Fig 4).

Likewise, the predictive levels of TG and NLR were significant at the assessment of the HFs severity value of 3 ($P <0.001$, AUC = 0.733, 95%CI:0.629–0.836; $P = 0.037$, AUC = 0.616, 95% CI:0.509–0.723, respectively) [Table 8]. Therefore, by examining the sensitivity and specificity of different points of TG and NLR at the HFs severity value of 3, we noticed that the best cut-off points for TG and NLR were as follows: 188.5 mg/dl with 71.4% sensitivity and 72% specificity for TG, and 1.45 with 62.9% sensitivity and 57% specificity for NLR (Fig 5).

## Discussion

The present study aimed to determine the association of some SIMs, LPs, and HFs in healthy menopausal women. Thus, we assessed the main obtained results. Menopause may activate IL-17 receptors on the surface of epithelial cells by lowering estrogen levels [16]. IL-17 can induce IL-6 and Matrix Metallo Proteinase (MMP) release by affecting endothelial cells, which results in reperfusion damage, thrombosis, and atherosclerosis [15]. Molnar and Waliullah et al. observed diminished estradiol and raised IL-17 levels in menopausal women [16,17]. Moreover, women with HFs may have lower levels of estradiol than women without HFs, according to Gast et al. [27]. Accordingly, we expected menopausal women with HFs to have lower

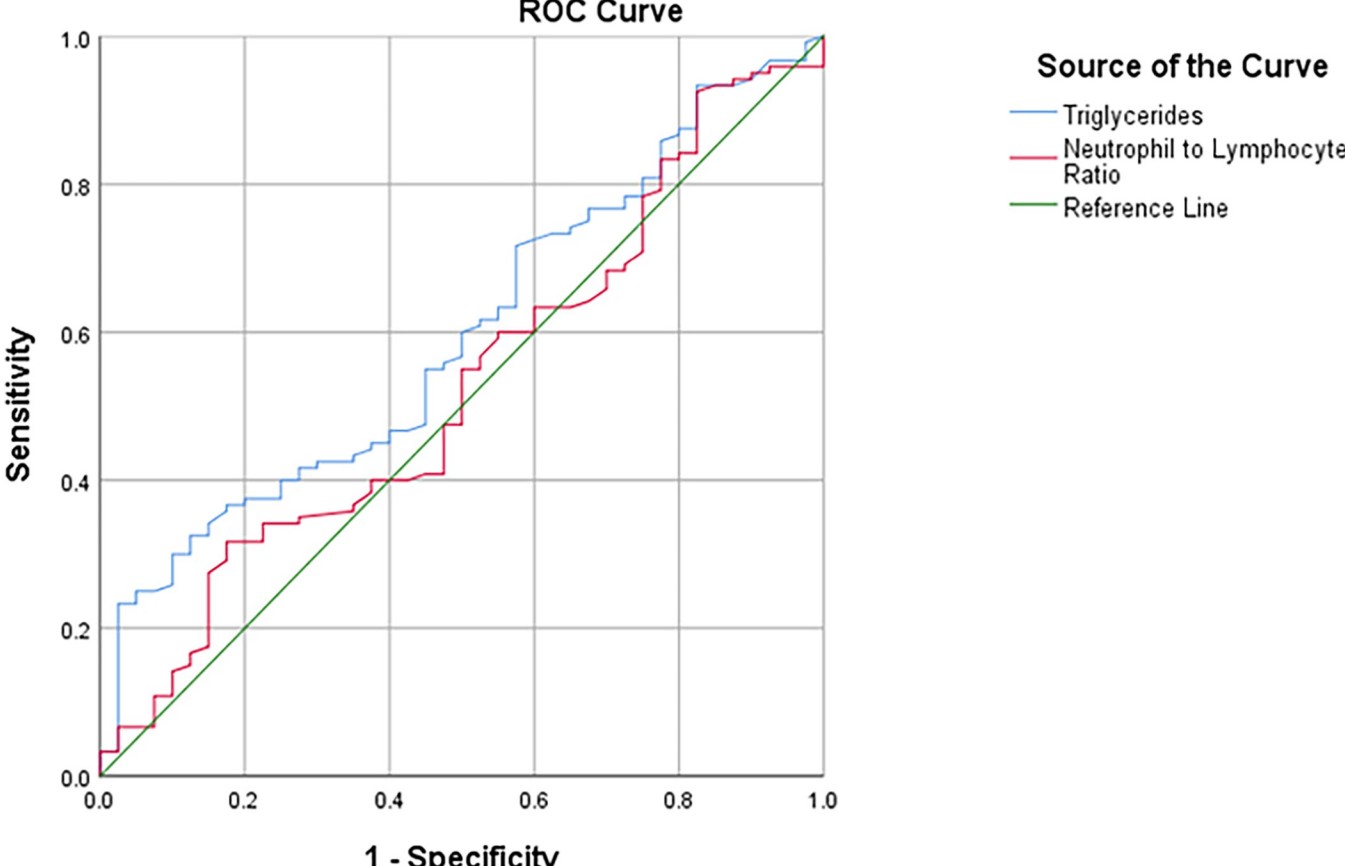

**Fig 3. ROC curve of TG (mg/dl) and NLR predictive levels to determine the prognosis of HFs severity 1.**

estradiol and higher IL-17 levels than menopausal women without HFs. We actually detected no significant association between IL-17 and HFs. However, compared to menopausal women without HFs, those with HFs tended to have lesser estradiol and increased IL-17 levels. In this study, we planned to gauge the serum IL-17 levels of menopausal women, while a simple expansion in circulating IL-17 may not be the reaction to estradiol depletion. Likewise, some studies noticed a rise in this cytokine in the synovial fluid of patients with rheumatoid arthritis [28]. Given the design of the study, we did not perform invasive procedures. Evaluating the local levels of IL-17 may lead to a better understanding of this issue. As per searching databases, only one study (Huang 2017) assessed the association of circulating IL-17 with HFs and uncovered no significant association [10]. Thus, it makes sense to do more research in this field.

Concerning the univariate analysis, the SWAN study ascertained a statistically significant association between hs-CRP and HFs, which was no more significant after controlling covariates. This point was probably a matter of shared risk factors, particularly overweight/obesity, the researchers stated [13]. Menopause can increase fat and accumulate abdominal adipose tissue in women directly or by declining estradiol, which may result in SIMs increase, including CRP, and thus, HFs occurrence [9]. Moreover, Tuomikoski et al. observed no association between hs-CRP changes and HFs [7]. Furthermore, CRP levels were not associated with total menopause rating scale (MRS) symptom score, Kaya et al. realized [29]. Since estradiol can be involved in CRP production in the liver [30], we determined estradiol as a likely confounding

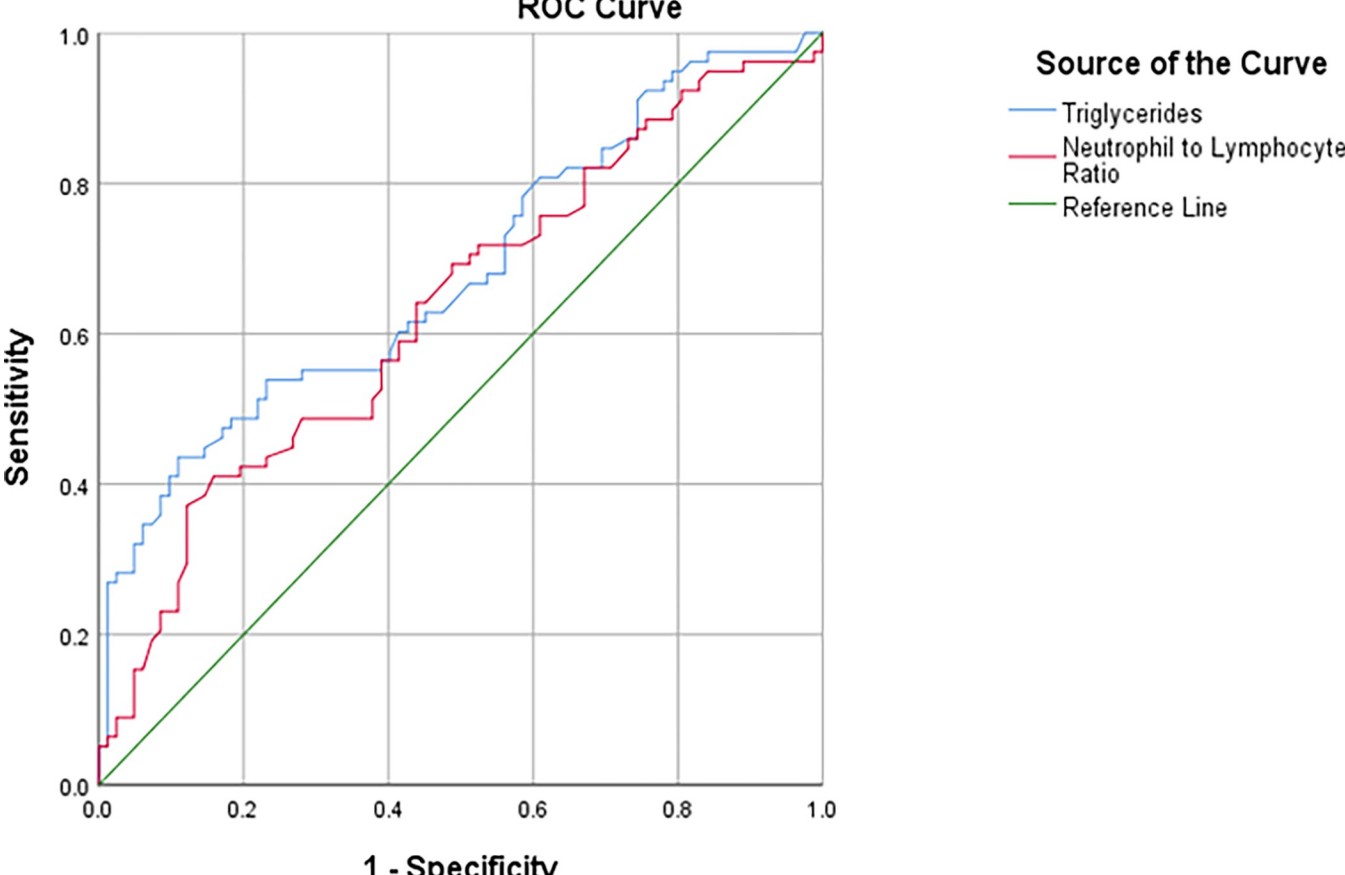

**Fig 4. ROC curve of TG (mg/dl) and NLR predictive levels to determine the prognosis of HFs severity 2.**

variable. Therefore, we compared estradiol levels in menopausal women without and with HFs to ensure their homogeneity. We observed no significant association between hs-CRP alterations and the HFs incidence and severity. These findings were in line with the SWAN study. Also, given the prevalence of overweight women in Guilan province, Iran, we measured BMI and WC/HC in both menopausal women without and with HFs to confirm their homogeneity in terms of these variables. In total, CRP may not have a simple increasing impact on HFs and further studies are needed in this field.

Chen et al. realized a drop in NLR after a decrease in neutrophil count and an increase in lymphocyte count in menopausal women. They ascribed this alteration to menopause-induced estradiol deficiency [20]. As stated before, some studies noted a connection between declined estradiol and extended IL-17 levels in menopausal women [16,17]. Tahmasebinia et al. also reported raised IL-17 levels associated with increased neutrophil counts [23]. Thus, as IL-17 enhanced, we expected neutrophil count and NLR to increase and estradiol to decrease. We found no significant association between NLR and HFs. However, we observed a tendency to expand IL-17 and increase the neutrophil count and NLR in menopausal women with HFs compared to those without HFs. Even though not significant, this finding was consistent with the pathophysiology of inflammation and dyslipidemia association with HFs [9]. Conversely, it contradicts Chen's data (dropped counts of neutrophils and NLR following decreased estradiol levels) [20]. We also observed that NLR significantly increased as the HFs severity

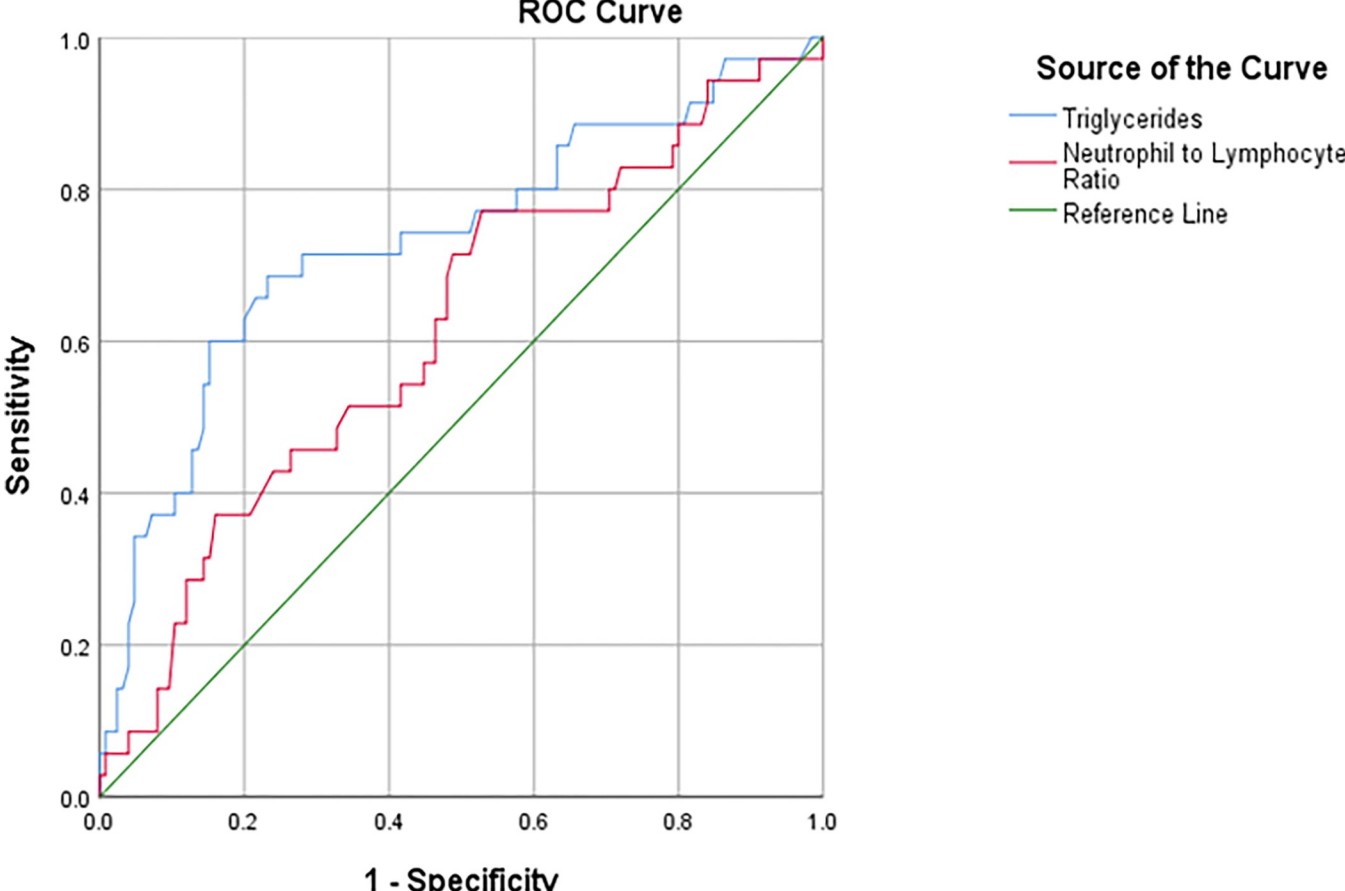

**Fig 5. ROC curve of TG (mg/dl) and NLR predictive levels to determine the prognosis of HFs severity 3.**

extended. Adjustment for the covariates still retained this association. Furthermore, Lee et al. noted increased LMR in menopause and considered it would be associated with decreased estradiol levels [24]. We observed women with HFs tended to expand LMR levels compared to those without HFs. However not statistically significant, this finding agreed with Lee's report and the pathophysiology of inflammation and dyslipidemia association with HFs [9,20]. We did not observe a significant association between the HFs severity and LMR.

The present study explored significant associations between some components of LPs and HFs. We found higher TG levels in menopausal women with HFs than those without HFs after the multivariate regression. As we observed, the menopausal women with HFs showed more elevated levels of TC, TG, and LDL-C than those without HFs. This increase, although not significant, was consistent with the pathophysiology of the relationship of dyslipidemia with HFs [9]. TC, LDL-C, and TC/HDL-C levels increased, and HDL-C levels diminished in menopausal women, as indicated by Taleb-Belkadi et al. These alterations were ascribed to estradiol deficiency in menopause [19]. Furthermore, we observed that HDL-C tended to increase and TC/HDL-C tended to decrease in the menopausal women with HFs, which was not consistent with the pathophysiology of the association of dyslipidemia with HFs [9]. Also, this finding was in contrast to those of the Taleb-Belkadi et al. study that reported TC/HDL-C levels increased, and HDL-C levels diminished in menopausal women [19]. Moreover, the levels of these TC and TG in menopausal women with HFs increased as HFs severity extended. This

finding could be according to the relationship between dyslipidemia and HFs [9]. After controlling the demographic and reproductive variables, we realized higher TG and HDL-C levels were associated with the HFs severity. The cumulative effect of HDL-C on the HFs severity is not justifiable. Therefore, further research is required to confirm such an association in menopausal women. Besides, Kaya et al. revealed that elevated TG levels were associated with higher total MRS symptom scores. However, they did not observe any significant difference in symptom severity-based groups regarding HDL-C levels [29]. Similarly, increased levels of TG were associated with higher total MRS scores in menopausal women, Cengiz et al. noticed [31].

We controlled the impact of physical activities and nutrition on participants. Some studies showed that more strenuous walking (such as Nordic walking) and walking duration might affect LPs [32]. We did not control the severity of walking in participants, which may be effective on LPs changes in menopausal women with HFs. As estradiol deficiency seems to be associated with dyslipidemia [9], we measured this sex hormone as a confounding variable. We did not observe any significant difference between the two groups of menopausal women without and with HFs in this regard.

The present study estimated the predictive levels of TG and NLR in the incidence and prognosis of various HFs severity levels, which was the first of its type to our knowledge. Relatedly, we noticed that with each 100 mg/dl increase in TG, the incidence of HFs can expand by nearly 1.5 times which can be practicable in the clinical approach.

In this study, hs-CRP negatively correlated with TG and TC/HDL-C in menopausal women with HFs, as we realized. Those correlations appear to be infirm. Moreover, they were not justified, given the pathophysiology of the association of inflammation and dyslipidemia with HFs [9]. Even though hs-CRP is an inflammatory marker, it sounds to require further study on menopause and its outcomes.

This study comprised some strengths: First, this study simultaneously assessed the relationship between SIMs, LPs, and the presence and severity of HFs. This is crucial because flushing is likely related to vascular dysfunction [9]. Therefore, investigating the alterations of SIMs and LPs in HFs severity-based groups also seems reasonable. Second, the participants with HFs experienced flushing for at least three months before enrollment and reported relatively reliable HFs frequency. Third, the participants only declared their daily HFs number, and we determined the HFs severity accordingly. Thus, we could prevent the participants from an unconscious bias concerning HFs severity report. Fourth, regarding low prices and ease of access, studies on NLR and LMR as hematologic inflammatory markers appear cost-effective.

This study included some limitations we need to address: First, structurally (cross-sectional study), it could not determine the causal relationship of SIMs and LPs with HFs. Second, we used the minimum sample size necessary to assess this relationship. As we noticed, the alterations of some variables, however not significant, were in line with the pathophysiology of the association between some SIMs, LPs, and HFs. Therefore, using a larger sample size may result in significant differences between groups of menopausal women. Third, in this study, we did not use a standard tool to evaluate food intake and physical activities in menopausal women. Instead, we asked them about their physical activities (type, frequency and duration of physical activity if done) and nutrition (frequency of having oily and fast foods if consumed). Fourth, participants recorded the number of their daily HFs subjectively. Accordingly, they may not have registered their accurate number of HFs.

## Conclusion

Concisely, we found that TG was related to HFs in menopausal women. Indeed, menopausal women with HFs had higher levels of TG than those without HFs. We also noticed a rise in

NLR, HDL-C, and TG levels as HFs severity enhanced. We furthermore noted negative correlations between hs-CRP and TG; and hs-CRP and TC/HDL-C in menopausal women with HFs. We also observed that TG and NLR could predict the incidence of HFs and determine the prognosis of the HFs severity values of 2 and 3. Hence, our study revealed that additive changes in some SIMs and LPs could be related to HFs and their severity. This event may suggest HFs as links between alterations in SIMs/LPs and their outcomes. Considering the interaction of these variables, surveying and focusing on SIMs' and LPs' levels in perimenopausal women may predict and decline the incidence and severity of HFs. Regarding the probable association of HFs with endothelial dysfunction [9], adjusting SIMs' and LPs' levels in perimenopausal women appears to prevent HFs and their potential outcomes through menopause. Moreover, researchers better perform longitudinal studies to elucidate the causal relationship among inflammation, dyslipidemia, and HFs.

## Supporting information

**S1 Fig. Comparison of the median of NLR among the HFs severity-based groups.**
(TIF)

**S2 Fig. Comparison of the mean of TC (mg/dl) among the HFs severity-based groups.**
(TIF)

**S3 Fig. Comparison of the median of TG (mg/dl) among the HFs severity-based groups.**
(TIF)

**S1 File. Data set.**
(PDF)

## Acknowledgments

We appreciate the Vice Chancellor of Research and Technology of Guilan University of Medical Sciences (GUMS), Rasht, Iran, for approval of this research project. We also value Hamzeh Yousefi Amin, the research laboratory supervisor, and his associates for their assistance and performance of the lab tests. Moreover, we would like to express our special thanks to all women who participated in this study.

## Author Contributions

**Conceptualization:** Nazila Didevar, Parvaneh Rezasoltani, Arash Pourgholaminejad, Ehsan Kazemnezhad Leyli, Tahereh Seyednoori, Ziba Zahiri Sorouri.

**Data curation:** Nazila Didevar, Ehsan Kazemnezhad Leyli.

**Formal analysis:** Nazila Didevar, Parvaneh Rezasoltani, Arash Pourgholaminejad, Ehsan Kazemnezhad Leyli.

**Investigation:** Nazila Didevar, Parvaneh Rezasoltani, Arash Pourgholaminejad.

**Methodology:** Nazila Didevar, Parvaneh Rezasoltani, Arash Pourgholaminejad, Ehsan Kazemnezhad Leyli.

**Project administration:** Nazila Didevar, Parvaneh Rezasoltani, Arash Pourgholaminejad, Ehsan Kazemnezhad Leyli.

**Resources:** Nazila Didevar, Parvaneh Rezasoltani, Arash Pourgholaminejad, Ehsan Kazemnezhad Leyli.

**Supervision:** Parvaneh Rezasoltani, Arash Pourgholaminejad.

**Validation:** Parvaneh Rezasoltani, Arash Pourgholaminejad, Ehsan Kazemnezhad Leyli, Tahereh Seyednoori.

**Visualization:** Nazila Didevar, Parvaneh Rezasoltani.

**Writing – original draft:** Nazila Didevar, Parvaneh Rezasoltani, Arash Pourgholaminejad.

**Writing – review & editing:** Nazila Didevar, Parvaneh Rezasoltani, Arash Pourgholaminejad.

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
