## [Decision Letter · Decision Letter 0]

30 Jan 2023

PONE-D-22-24749Interleukin-17, C-Reactive Protein, Neutrophil-to-Lymphocyte ratio, Lymphocyte-to-Monocyte ratio, and lipid profiles in healthy menopausal women with or without hot flashes: A cross-sectional studyPLOS ONE

Dear Dr. Rezasoltani,

Thank you for submitting your manuscript to PLOS ONE. After careful consideration, we feel that it has merit but does not fully meet PLOS ONE’s publication criteria as it currently stands. Therefore, we invite you to submit a revised version of the manuscript that addresses the points raised during the review process.

The authors are encouraged to write more clearly so that an international readership can understand the main issues addressed in this important paper. Below are a few comments the authors need to address in addition to those of the anonymous reviewers.

**Materials and Methods - **Page 7; Lines 149 – 153: The authors wrote, “Based on Fig 1 a total of 1987 women were **interviewed**, and 1816 of them were excluded from the study in terms of the lack of inclusion criteria. There were 171 eligible women, 11 of whom were unwilling to continue the cooperation, and a total of 160 were recruited. According to their response to HFs status, participating women were divided into two groups without HFs (N꞊40) and with HFs (N꞊120).”

It is unclear what kind of interview was conducted on the initial sample of 1987.

I think it flows better if the authors state that the women did not meet the inclusion criteria, rather than state that they lacked the inclusion criteria.

It is quite interesting that the process of selection of the eligible sample led to the exact number of women calculated as the minimum sample size. Is this approach reproducible by other researchers? Did the authors stop recruiting the moment they attained the minimum sample size, or did recruitment continue until the end of that work day? What steps did the researchers put in place to avoid selection bias?

Page 7; line 154 – 155: The authors wrote, “...were given a card to record their HFs number for two weeks in a pilot manner.” Authors should edit to read, “...were given a card to record the number of HFs they experienced over a two-week period as part of a pilot study.” PLoS One does not edit accepted manuscripts. I see that the authors have already employed the services of a professional English language editor, but more editing is required to improve the readability of the mannuscript.

**Results** - Figure 1: The authors state that 1816 women were excluded in the first stage but only 1716 women were accounted for. Why were the other 100 women excluded? These 100 women need to be reflected in Figure 1.

**Discussion** - This section needs to be more concisely written as it is currently difficult to read and understand. Are there any recommendations for clinical practice, based on the findings of this study?

We look forward to receiving your revised manuscript.

Kind regards,

Funmilola M. OlaOlorun, PhD

Academic Editor

PLOS ONE

Journal Requirements:

Reviewers' comments:

Reviewer's Responses to Questions

**Comments to the Author**

1. Is the manuscript technically sound, and do the data support the conclusions?

Reviewer #1: No

Reviewer #2: Yes

2. Has the statistical analysis been performed appropriately and rigorously? 

Reviewer #1: No

Reviewer #2: Yes

3. Have the authors made all data underlying the findings in their manuscript fully available?

Reviewer #1: Yes

Reviewer #2: Yes

4. Is the manuscript presented in an intelligible fashion and written in standard English?

Reviewer #1: Yes

Reviewer #2: Yes

5. Review Comments to the Author

Reviewer #1: # Review

- Lines 43-46: The following statement in the Abstract section should be written more clearly since not all those markers were measured in serum samples and even more, some of them were calculated: „The serum levels of Interleukin-17 (IL-17), C-Reactive Protein (CRP), Neutrophil-to-Lymphocyte Ratio (NLR), Lymphocyte-to-Monocyte Ratio (LMR), Total Cholesterol (TC), Triglycerides (TG), Low-Density Lipoprotein Cholesterol (LDL-C), High-Density Lipoprotein Cholesterol (HDL-C), and TC/HDL-C were measured in each group“.

- Did the Authors exclude women with CRP ≥10 mg/L in order to exclude acute inflammation?

- In the Statistical analysis subsection it should be mentioned that continuous data with non-normal distribution were presented as median (IQ range).

- The same results should be presented either in tables or in figures, but not both (e.g. Table 4 and Figures 3, 4).

- Why did not the Authors include IL-17 in the correlation analysis (Table 7)?

- The Discussion section should be shortened in order to be more clear, coherent and concise.

- Lines 470-471: the Authors are not cited properly (i.e. reference 23): „Moreover, Pourgholaminejad et al. reported an increase in 471 the neutrophil count with an increase in IL-17 [23]“.

- Lines 488-490: the Authors are not cited properly (i.e. reference 19): „Belkadi et al. observed a 489 raise in TC, LDL-C and TC/HDL-C and a drop in HDL-C in menopausal women and 490 attributed these alterations to estradiol deficiency in menopausal women [19]“.

- Line 534: the following statement should be written to be more clear: „Study samples may not have reported their exact HFs number“.

- Line 536: the following statement should be written to be more clear: „HFs had a relationship with TG“.

- The units are missing in some figures.

Reviewer #2: Dear Editor

This is a good manuscript “Interleukin-17, C-Reactive Protein, Neutrophil-to-Lymphocyte ratio, Lymphocyte-to- Monocyte ratio, and lipid profiles in healthy menopausal women with or without hot flashes: A cross-sectional study”. The subject of the manuscript is fully consistent with the aims and scope of the journal « Plos one». The research methodology is fully consistent with the aims declared by the authors. Their conclusions are also consistent with the set goals, however, some issues need to be reconsidered:

- Please explain all abbreviations in the abstract and manuscript.

Abstracts

1) Abstract should be informative, I think abstract need to be rewrite to be more easy to get!

2) Keywords: are these keywords are Mesh terms? Word that serves as a keyword, as to the meaning of that condition must be a Mesh term

Introduction

- There are some sentences that are difficult to understand, and the paper needs an English reviewer. Please edit.

-introduction is too long I think it would be possible to summarize some parts

Methods

- I couldn’t find any methodological tools and definition for evaluation the food intake in these patients as authors know nutritional intake and status play a very important role in post-menopausal women’s health and it should be evaluated as a very critical confounder in this relationship.

- What about physical activity, I also couldn’t find in the method part any explanation for evaluation of physical activity of participants.

Results

-Evaluation of inflammatory markers especially Interleukins and CRP are definitely related to the time of a day, what is the exact time of sampling and are there any planned and schedule for this?

Discussion

- The strength and limitations of the study should be extensively described. For example limited number of sample size and also do not evaluating the very important confounders such as food intake and physical activity should be described as limitation.

-

6. PLOS authors have the option to publish the peer review history of their article (what does this mean?). If published, this will include your full peer review and any attached files.

Reviewer #1: No

Reviewer #2: No

---

## [Author Response · Author response to Decision Letter 0]

4 Apr 2023

March 14, 2023

Dear Dr. Emily Chenette

Editor in Chief

PLOS ONE

Research article

Interleukin-17, C-Reactive Protein, Neutrophil-to-Lymphocyte ratio, Lymphocyte-to-Monocyte ratio, and lipid profiles in healthy menopausal women with or without hot flashes: A cross-sectional study (PONE-D-22-24749) 

Nazila Didevar, Parvaneh Rezasoltani, Arash Pourgholaminejad, Ehsan Kazemnezhad Leyli, Tahereh Seyednoori, Ziba Zahiri Sorouri

Thank you for your email (Jan 30, 2023) about our joint manuscript cited above. We appreciate your reviewers' valuable remarks and suggestions that have improved our manuscript. The revisions applied are as follows.

I value your kind and quick response.

Yours’ Sincerely, 

Parvaneh Rezasoltani, PhD

Corresponding Author

Assistant Professor, Department of Midwifery, School of Nursing and Midwifery, Guilan University of Medical Sciences, Rasht, Iran

rezasoltani@gums.ac.ir; rezasoltani49@gmail.com

Editors’ comments

 Academic Editor

- The authors are encouraged to write more clearly so that an international readership can understand the main issues addressed in this important paper. 

Authors’ reply: With respect and gratitude, we edited the article appropriately to clarify the content.

- Materials and Methods - Page 7; Lines 149 – 153: The authors wrote, “Based on Fig 1 a total of 1987 women were interviewed, and 1816 of them were excluded from the study in terms of the lack of inclusion criteria. There were 171 eligible women, 11 of whom were unwilling to continue the cooperation, and a total of 160 were recruited. According to their response to HFs status, participating women were divided into two groups without HFs (N꞊40) and with HFs (N꞊120).” It is unclear what kind of interview was conducted on the initial sample of 1987.

Authors’ reply: In this study, we interviewed all the menopausal women individually and asked them the same questions in a cosecutive manner, according to the inclusion criteria.

- I think it flows better if the authors state that the women did not meet the inclusion criteria, rather than state that they lacked the inclusion criteria.

Authors’ reply: We appreciate your recommendation. Therefore, concerning your two latter suggestions, we altered the above section to “Based on Fig 1, the first author (ND) individually interviewed a total of 1987 women in a consecutive manner, according to the inclusion criteria. As the sampling proceeded, she excluded 1816 women from the study because they did not meet the inclusion criteria. Finally, she collected 171 eligible women, out of which 11 were unwilling to continue the cooperation. Therefore, a total of 160 women were enrolled in the study. According to their response to HFs status, the participating women were divided into two groups without HFs (N꞊40) and with HFs (N꞊120).” (Page 7; Lines 153-159)

- It is quite interesting that the process of selection of the eligible sample led to the exact number of women calculated as the minimum sample size. Is this approach reproducible by other researchers? Did the authors stop recruiting the moment they attained the minimum sample size, or did recruitment continue until the end of that work day? What steps did the researchers put in place to avoid selection bias?

Authors’ reply: We value your consideration. We emphasize that menopausal women with different statuses might have visited the hospital on various days. We recruited them on definite days randomly using consecutive sampling. In other words, we enlisted eligible menopausal women who visited the hospital randomly only based on meeting the inclusion criteria. Thus, the researchers had no role in their selection. Therefore, other researchers will be able to apply such a method, but due to the randomness of the women included in the study, they may obtain different results. Furthermore, we mention that we continued recruiting women until the end of the workday we achieved the total sample size. Moreover, we note that we used a consecutive method for sampling. Therefore, we could replace the excluded women with others as the sampling proceeded. In this way, we attained the estimated sample size. Thus, we revised the related part as: “Finally, she collected 171 eligible women, out of which 11 were unwilling to continue the cooperation. Therefore, a total of 160 women were enrolled in the study.” (Page 7; Lines 156-157)

- Page 7; line 154 – 155: The authors wrote, “...were given a card to record their HFs number for two weeks in a pilot manner.” Authors should edit to read, “... were given a card to record the number of HFs they experienced over a two-week period as part of a pilot study.” 

Authors’ reply: Thank you very much for your advice. We revised the sentence as you suggested: “ The women with HFs, who had experienced HFs during the three months before enrollment, were given a card and asked to record the number of HFs they experienced over a two-week period as part of a pilot study.” (Page 7; Lines 160-161)

- Results - Figure 1: The authors state that 1816 women were excluded in the first stage but only 1716 women were accounted for. Why were the other 100 women excluded? These 100 women need to be reflected in Figure 1.

Authors’ reply: With respect and expressions of regret, the mentioned point was only a typographical mistake. Reconsidering the data, we noticed that the number of women excluded from the study due to diabetes was 241, of which 141 were mistyped. We have corrected the mentioned point with many thanks for your consideration (Page 8; Fig. 1).

- Discussion - This section needs to be more concisely written as it is currently difficult to read and understand. Are there any recommendations for clinical practice, based on the findings of this study?

Authors’ reply: We would appreciate informing you that we rewrote the discussion section and made it more concise. We also added a few suggestions for a clinical approach in this regard.

Reviewer #1

- Lines 43-46: The following statement in the Abstract section should be written more clearly since not all those markers were measured in serum samples and even more, some of them were calculated: „The serum levels of Interleukin-17 (IL-17), C-Reactive Protein (CRP), Neutrophil-to-Lymphocyte Ratio (NLR), Lymphocyte-to-Monocyte Ratio (LMR), Total Cholesterol (TC), Triglycerides (TG), Low-Density Lipoprotein Cholesterol (LDL-C), High-Density Lipoprotein Cholesterol (HDL-C), and TC/HDL-C were measured in each group“.

Authors’ reply: With many thanks and respect, we inform you that based on your comment, we rewrote and modified the mentioned point as follows: “In addition to clinical variables and HFs experience, we measured the fasting serum levels of SIMs and lipid profiles (LPs), including Interleukin-17 (IL-17), high-sensitivity C-Reactive Protein (hs-CRP), Total Cholesterol (TC), Triglycerides (TG), Low-Density Lipoprotein Cholesterol (LDL-C), and High-Density Lipoprotein Cholesterol (HDL-C) in each group. Then, we calculated TC/HDL-C concerning the related variables and determined Neutrophil-to-Lymphocyte Ratio (NLR), and Lymphocyte-to-Monocyte Ratio (LMR), according to Complete Blood Count (CBC) quantitative parameters in each group.” (Page 2; Lines 43-50)

- Did the Authors exclude women with CRP ≥10 mg/L in order to exclude acute inflammation?

Authors’ reply: Many thanks for your notice. Referring to our data, we observed that there were only two menopausal women (one in each severity value of 1 and 2) who had hs-CRP levels≥10 mg/L. None of them had clinical symptoms of inflammation. Moreover, they stated no disease history to put them in the exclusion criteria.

- In the Statistical analysis subsection it should be mentioned that continuous data with non-normal distribution were presented as median (IQ range).

Authors’ reply: With gratitude for your consideration, we included the suggested item in such a way: “The continuous variables with normal distribution were presented as the mean [standard deviation (SD)], those with non-normal distribution as the median (IQ range), and the categorical variables as frequency (percentage).” (Page 11; Lines 250-252)

- The same results should be presented either in tables or in figures, but not both (e.g. Table 4 and Figures 3, 4).

Authors’ reply: Thank you very much for your recommendation. We revised the content based on your comment (Page 13; Line 290 / Page 14; Line 309 / Page 15; Line 344 / Page 16; Line 349).

- Why did not the Authors include IL-17 in the correlation analysis (Table 7)?

Authors’ reply: With many thanks for your consideration, we declare that the mentioned point was a typographical mistake. We added the missing IL-17 to the correlation table, as below: (Page 21, Table 7)

TC/HDL-C LDL-C HDL-C TG TC 

0.074 -0.123 -0.110 0.141 0.038 Correlation coefficient IL-17

0.423 0.181 0.230 0.125 0.677 Sig. (2-tailed) 

120 120 120 120 120 N 

- The Discussion section should be shortened in order to be more clear, coherent and concise.

Authors’ reply: We would appreciate the opportunity to implement your advice. Therefore, we rewrote and modified the discussion section to be clear and more concise.

- Lines 470-471: the Authors are not cited properly (i.e. reference 23): „Moreover, Pourgholaminejad et al. reported an increase in the neutrophil count with an increase in IL-17 [23]“.

Authors’ reply: Below, we appreciatively corrected the citation: “Tahmasebinia et al. also reported raised IL-17 levels associated with increased neutrophil counts [23].” (Page 24; Lines 480-481)

- Lines 488-490: the Authors are not cited properly (i.e. reference 19): „Belkadi et al. observed a raise in TC, LDL-C and TC/HDL-C and a drop in HDL-C in menopausal women and attributed these alterations to estradiol deficiency in menopausal women [19]“.

Authors’ reply: Many thanks for your attentive remark. We adjusted the citation as follows: “TC, LDL-C, and TC/HDL-C levels increased, and HDL-C levels diminished in menopausal women, as indicated by Taleb-Belkadi et al. These alterations were ascribed to estradiol deficiency in menopause [19].” (Page 25; Lines 501-503)

- Line 534: the following statement should be written to be more clear: „Study samples may not have reported their exact HFs number“.

Authors’ reply: Based on your careful remark, we corrected this sentence to be more intelligible: “Participants recorded the number of their daily HFs subjectively. Accordingly, they may not have registered their accurate number of HFs.” (Page 27; Lines 552-553)

- Line 536: the following statement should be written to be more clear: „HFs had a relationship with TG“.

Authors’ reply: We gratefully revised this sentence to be more comprehensible, according to your comment: “Concisely, we found that TG was related to HFs in menopausal women. Indeed, menopausal women with HFs had higher levels of TG than those without HFs.” (Page 27; Lines 555-556)

- The units are missing in some figures.

Authors’ reply: We state that we corrected all missing units in the titles of the figures with special thanks for your carefulness (Page 21; Lines 419 and 422 / Page 22; Lines 425, 428, 432, and 435).

Reviewer #2

- Please explain all abbreviations in the abstract and manuscript.

Authors’ reply: According to your comment, we explained all abbreviations in both the abstract and manuscript.

Abstract

1) Abstract should be informative, I think abstract need to be rewrite to be more easy to get!

Authors’ reply: With thanks and respect, we rewrote the abstract to be more intelligible (Pages 2-3).

2) Keywords: are these keywords are Mesh terms? Word that serves as a keyword, as to the meaning of that condition must be a Mesh term.

Authors’ reply: Much obliged for your attentive comment; we state that all the keywords are MeSH terms. While designing the proposal for this study, we checked the keywords through PubMed and MeSH databases according to headings.

Introduction

- There are some sentences that are difficult to understand, and the paper needs an English reviewer. Please edit.

- introduction is too long I think it would be possible to summarize some parts.

Authors’ reply: Thank you very much for your notice. According to your comment, we edited this part.

Methods

- I couldn’t find any methodological tools and definition for evaluation the food intake in these patients as authors know nutritional intake and status play a very important role in post-menopausal women’s health and it should be evaluated as a very critical confounder in this relationship.

Authors’ reply: We thankfully mention that we did not use any methodological tools to assess nutrition. Instead, we asked some questions about nutritional status in a questionnaire, the validity and reliability of which were verified by ten faculty members of the School of Nursing and Midwifery of Guilan Medical Sciences. These questions were as shown: “Do you ever consume fatty and fast food? If so, how often do you use it?” Therefore, according to your suggestion, we put the non-use of tools in the analysis of nutritional status in the limitations so that researchers in the future can use the related tools to evaluate the exact effects of nutrition in this study (Page 27; Lines 548-551).

- What about physical activity, I also couldn’t find in the method part any explanation for evaluation of physical activity of participants.

Authors’ reply: As I said about nutrition, we did not use a methodological tool for physical activities either. We asked valid and reliable questions verified by ten faculty members of the School of Nursing and Midwifery of Guilan University of Medical Sciences. These questions (in the demographic and reproductive form) were as follows: “Do you ever work out? If so, what kind of activity, how many times a week, and how long (in minutes) each time?” (Pages 17-18; Tables 1 and 2).

Results

- Evaluation of inflammatory markers especially Interleukins and CRP are definitely related to the time of a day, what is the exact time of sampling and are there any planned and schedule for this?

Authors’ reply: All participants had to take a blood test between 8 and 10 am, concerning the laboratory regulations. Therefore, we planned a schedule according to which participants could refer to the research laboratory and take the blood test (Page 9; Line 210).

Discussion

- The strength and limitations of the study should be extensively described. For example limited number of sample size and also do not evaluating the very important confounders such as food intake and physical activity should be described as limitation.

Authors’ reply: Thank you very much for your consideration. Based on your comments, we explained this part more extensively (Page 27; Lines 544-551).

---

## [Decision Letter · Decision Letter 1]

4 Jul 2023

PONE-D-22-24749R1Interleukin-17, C-Reactive Protein, Neutrophil-to-Lymphocyte ratio, Lymphocyte-to-Monocyte ratio, and lipid profiles in healthy menopausal women with or without hot flashes: A cross-sectional studyPLOS ONE

Dear Dr. Rezasoltani,

Thank you for submitting your manuscript to PLOS ONE. After careful consideration, we feel that it has merit but does not fully meet PLOS ONE’s publication criteria as it currently stands. Therefore, we invite you to submit a revised version of the manuscript that addresses the points raised during the review process.

Thank you for making the requested edits to improve the quality of your manuscript. One of the reviewers would like you to include relevant information from two papers as indicated in their review. Kindly address this request.==============================

We look forward to receiving your revised manuscript.

Kind regards,

Funmilola M. OlaOlorun, PhD

Academic Editor

PLOS ONE

Journal Requirements:

Reviewers' comments:

Reviewer's Responses to Questions

**Comments to the Author**

1. If the authors have adequately addressed your comments raised in a previous round of review and you feel that this manuscript is now acceptable for publication, you may indicate that here to bypass the “Comments to the Author” section, enter your conflict of interest statement in the “Confidential to Editor” section, and submit your "Accept" recommendation.

Reviewer #2: All comments have been addressed

Reviewer #3: All comments have been addressed

2. Is the manuscript technically sound, and do the data support the conclusions?

Reviewer #2: (No Response)

Reviewer #3: Yes

3. Has the statistical analysis been performed appropriately and rigorously? 

Reviewer #2: (No Response)

Reviewer #3: Yes

4. Have the authors made all data underlying the findings in their manuscript fully available?

Reviewer #2: (No Response)

Reviewer #3: Yes

5. Is the manuscript presented in an intelligible fashion and written in standard English?

Reviewer #2: (No Response)

Reviewer #3: Yes

6. Review Comments to the Author

Reviewer #2: all the comments has been fully addressed and The revision was made, and I was satisfied with the response.

Reviewer #3: The paper is well revised with a clear language. The topic is important for the region where the study took place. Before proceed authors should briefly discuss recent studies PMID: 30744450 PMID: 28472898

7. PLOS authors have the option to publish the peer review history of their article (what does this mean?). If published, this will include your full peer review and any attached files.

Reviewer #2: No

Reviewer #3: No

---

## [Author Response · Author response to Decision Letter 1]

3 Aug 2023

July 27, 2023

Dear Dr. Emily Chenette, PhD

Editor in Chief, PLOS ONE

Research article

Interleukin-17, C-Reactive Protein, Neutrophil-to-Lymphocyte ratio, Lymphocyte-to-Monocyte ratio, and lipid profiles in healthy menopausal women with or without hot flashes: A cross-sectional study (PONE-D-22-24749R1) 

Nazila Didevar, Parvaneh Rezasoltani, Arash Pourgholaminejad, Ehsan Kazemnezhad Leyli, Tahereh Seyednoori, Ziba Zahiri Sorouri

Thank you for your email (July 5, 2023) about our joint manuscript cited above. We appreciate your reviewers' valuable remarks and suggestions that have improved our manuscript. The revisions applied are as follows.

I value your kind and quick response.

Sincerely yours

Parvaneh Rezasoltani, PhD

Corresponding Author

Assistant Professor, Department of Midwifery, School of Nursing and Midwifery, Guilan University of Medical Sciences, Rasht, Iran

rezasoltani@gums.ac.ir; rezasoltani49@gmail.com

Editors’ comments

Reviewer #2

- All the comments has been fully addressed and the revision was made, and I was satisfied with the response.

Authors’ reply: We gratefully express our special thanks for your consideration and valuable comments, which helped a lot our manuscript to be improved.

Reviewer #3

- The paper is well revised with a clear language. The topic is important for the region where the study took place. Before proceed authors should briefly discuss recent studies PMID: 30744450 PMID: 28472898

Authors’ reply: We would appreciate the opportunity to implement your guidance. Therefore, we briefly cited the related key features of the mentioned papers in the appropriate sections:

- “Furthermore, CRP levels were not associated with total menopause rating scale (MRS) symptom score, Kaya et al. realized [29].” (Page 23; Lines 468-469)

- “Besides, Kaya et al. revealed that elevated TG levels were associated with higher total MRS symptom scores. However, they did not observe any significant difference in symptom severity-based groups regarding HDL-C levels [29]. Similarly, increased levels of TG were associated with higher total MRS scores in menopausal women, Cengiz et al. noticed [31].” (Pages 25-26; Lines 516-521)

Journal Requirements:

Authors’ reply: With many thanks, we let you know that we rechecked the list of references and made sure they were correct and accurate. Moreover, we state that no retracted papers were among those we used. Also, if available, we added their PMID to the list of references and wrote down the full title of the journals in citations. We also mention that according to the suggestion of one of the respected reviewers, we added two papers to the previous list of references, which include numbers 29 and 31. The updated list is as follows (Pages 30-34; Lines 618- 762):

References

1. Shifren JL. Schiff I. Menopause. In: Berek JS. Berek & Novak’s Gynecology Essentials: Wolters Kluwer Health; 2020. p. 934.

2. Casper RF. Clinical Manifestations and Diagnosis of Menopause; 2023 [cited 2023 Jun 29]. Database: UpToDate [Internet]. Available from: https://www.uptodate.com/contents/clinical-manifestations-and-diagnosis-of-menopause

3. Santen RJ, Loprinzi CL, Casper RF. Menopausal Hot Flashes; 2023 [cited 2020 Apr 27]. Database: UpToDate [Internet]. Available from: https://www.uptodate.com/contents/menopausal-hot-flashes

4. Thurston RC, Joffe H. Vasomotor symptoms and menopause: findings from the Study of Women's Health across the Nation. Obstetrics and Gynecology Clinics of North America. 2011;38(3):489-501. doi: 10.1016/j.ogc.2011.05.006 PMID: 21961716

5. Franco OH, Muka T, Colpani V, Kunutsor S, Chowdhury S, Chowdhury R, et al. Vasomotor symptoms in women and cardiovascular risk markers: Systematic review and meta-analysis. Maturitas. 2015;81(3):353-361. doi: 10.1016/j.maturitas.2015.04.016 PMID: 26022385

6. Thurston RC, El Khoudary SR, Sutton-Tyrrell K, Crandall CJ, Gold EB, Sternfeld B, et al. Vasomotor symptoms and lipid profiles in women transitioning through menopause. Obstetrics and Gynecology. 2012;119(4):753-761. doi: 10.1097/AOG.0b013e31824a09ec PMID: 22433339

7. Tuomikoski P, Mikkola TS, Hämäläinen E, Tikkanen MJ, Turpeinen U, Ylikorkala O. Biochemical markers for cardiovascular disease in recently postmenopausal women with or without hot flashes. Menopause. 2010;17(1)145-151.doi: 10.1097/gme.0b013e3181acefd5 PMID: 19602991

8. Muka T, Imo D, Jaspers L, Colpani V, Chaker L, van der Lee SJ, et al. The global impact of non-communicable diseases on healthcare spending and national income: a systematic review. European Journal of Epidemiology. 2015;30(4):251-277. doi: 10.1007/s10654-014-9984-2 PMID: 25595318

9. Muka T. Women' s health: Implications of diet and cardiometabolic risk factors. Doctoral dissertation, the Netherlands, the Erasmus University Rotterdam. 2016. Available from: https://repub.eur.nl/pub/93261/

10. Huang WY, Hsin IL, Chen DR, Chang CC, Kor CT, Chen TY, et al. Circulating interleukin-8 and tumor necrosis factor-alpha are associated with hot flashes in healthy postmenopausal women. PLoS One. 2017;12(8):e0184011. doi: 10.1371/journal.pone.0184011 PMID: 28846735

11. Chedraui P, Jaramillo W, Pérez-López FR, Escobar GS, Morocho N, Hidalgo L. Pro-inflammatory cytokine levels in postmenopausal women with the metabolic syndrome. Gynecological Endocrinology. 2011;27(9):685-691. doi: 10.3109/09513590.2010.521270 PMID: 20937002

12. Karaoulanis SE, Daponte A, Rizouli KA, Rizoulis AA, Lialios GA, Theodoridou CT, et al. The role of cytokines and hot flashes in perimenopausal depression. Annals of General Psychiatry. 2012;11:9. doi: 10.1186/1744-859X-11-9 PMID: 22490187

13. Thurston RC, El Khoudary SR, Sutton-Tyrrell K, Crandall CJ, Gold E, Sternfeld B, et al. Are vasomotor symptoms associated with alterations in hemostatic and inflammatory markers? Findings from the Study of Women's Health Across the Nation. Menopause. 2011;18(10):1044-1051. doi: 10.1097/gme.0b013e31821f5d39 PMID: 21926929

14. Malutan AM, Dan M, Nicolae C, Carmen M. Proinflammatory and anti- inflammatory cytokine changes related to menopause. Menopause review/Przegląd Menopauzalny. 2014;13(3):162-168. doi: 10.5114/pm.2014.43818 PMID: 26327849

15. Miossec P, Korn T, Kuchroo VK. Interleukin-17 and type 17 helper T cells. The New England Journal of Medicine. 2009;361(9):888-898. doi: 10.1056/NEJMra0707449 PMID: 19710487

16. Molnár I, Bohaty I, Somogyiné-Vári É. High prevalence of increased interleukin-17A serum levels in postmenopausal estrogen deficiency. Menopause. 2014;21(7):749- 752. doi: 10.1097/GME.0000000000000125 PMID: 24253487

17. Waliullah S, Sharma V, Srivastava RN, Pradeep Y, Mahdi AA, Kumar S. IL-17A and IL-23 cytokines and their relation with estrogen in post menopausal osteoporosis. International Journal of Orthopaedics Sciences. 2018;4(1):864-866. doi: 10.22271/ortho.2018.v4.i1m.124

18. Joffe HV, Ridker PM, Manson JE, Cook NR, Buring JE, Rexrode KM. Sex hormone-binding globulin and serum testosterone are inversely associated with C-reactive protein levels in postmenopausal women at high risk for cardiovascular disease. Annals of Epidemiology. 2006;16(2):105-112. doi: 10.1016/j.annepidem.2005.07.055 PMID: 16216530

19. Taleb-Belkadi O, Chaib H, Zemour L, Fatah A, Chafi B, Mekki K. Lipid profile, inflammation, and oxidative status in peri- and postmenopausal women. Gynecological Endocrinology. 2016;32(12):982-985. doi: 10.1080/09513590.2016.1214257 PMID: 27558905

20. Chen Y, Zhang Y, Zhao G, Chen C, Yang P, Ye S, et al. Difference in leukocyte composition between women before and after menopausal age, and distinct sexual dimorphism. PLoS One. 2016;11(9):e0162953. doi: 10.1371/journal.pone.0162953 PMID: 27657912

21. Pourgholaminejad A, Tahmasebinia F. The role of Th17 cells in immunopathogenesis of neuroinflammatory disorders. Neuroimmune Diseases: Springer. 2019. p. 83-107. doi: 10.1007/978-3-030-19515-1_3

22. Flannigan KL, Ngo VL, Geem D, Harusato A, Hirota SA, Parkos CA, et al. IL-17A-mediated neutrophil recruitment limits expansion of segmented filamentous bacteria. Mucosal Immunology. 2017;10(3):673-684. doi: 10.1038/mi.2016.80 PMID: 27624780

23. Tahmasebinia F,Pourgholaminejad A. The role of Th17 cells in auto-inflammatory neurological disorders. Progress in Neuropsychopharmacology and Biological Psychiatry. 2017;79(Pt B):408-416. doi: 10.1016/j.pnpbp.2017.07.023 PMID: 28760387

24. Lee JS, Kim NY, Na SH, Youn YH, Shin CS. Reference values of neutrophil-lymphocyte ratio, lymphocyte-monocyte ratio, platelet-lymphocyte ratio, and mean platelet volume in healthy adults in South Korea. Medicine. 2018;97(26):e11138. doi: 10.1097/MD.0000000000011138 PMID: 29952958

25. Harlow SD, Gass M, Hall JE, Lobo R, Maki P, Rebar RW, et al. Executive summary of the Stages of Reproductive Aging Workshop + 10: addressing the unfinished agenda of staging reproductive aging. Menopause. 2012;19(4):387-395. doi: 10.1097/gme.0b013e31824d8f40 PMID: 22344196

26. Tao M, Shao H, Li C, Teng Y. Correlation between the modified Kupperman Index and the Menopause Rating Scale in Chinese women. Patient Preference and Adherence. 2013;7:223-229. doi: 10.2147/PPA.S42852 PMID: 23569361

27. Gast G-CM, Samsioe GN, Grobbee DE, Nilsson PM, van der Schouw YT. Vasomotor symptoms, estradiol levels and cardiovascular risk profile in women. Maturitas. 2010;66(3):285-290. doi: 10.1016/j.maturitas.2010.03.015 PMID: 20400247

28. Gaffen SL. The role of interleukin-17 in the pathogenesis of rheumatoid arthritis. Current Rheumatology Reports. 2009;11(5):365-370. doi: 10.1007/s11926-009-0052-y PMID: 19772832

29. Kaya C, Cengiz H, Yeşil A, Ekin M, Yaşar L. The relation among steroid hormone levels, lipid profile and menopausal symptom severity. Journal of Psychosomatic Obstetrics & Gynecology. 2017;38(4):284-291. doi: 10.1080/0167482X.2017.1321633 PMID: 28472898

30. Davison S, Davis SR. New markers for cardiovascular disease risk in women: impact of endogenous estrogen status and exogenous postmenopausal hormone therapy. The Journal of Clinical Endocrinology & Metabolism. 2003;88(6):2470 -2478. doi: 10.1210/jc.2002-021929 PMID: 12788842

31. Cengiz H, Kaya C, Suzen Caypinar S, Alay I. The relationship between menopausal symptoms and metabolic syndrome in postmenopausal women. Journal of Obstetrics and Gynaecology. 2019;39(4):529-533. doi:10.1080/01443615.2018.1534812 PMID: 30744450 

32. Song M-S, Yoo Y-K, Choi C-H, Kim N-C. Effects of nordic walking on body composition, muscle strength, and lipid profile in elderly women. Asian Nursing Research. 2013;7(1):1-7. doi: 10.1016/j.anr.2012.11.001 PMID: 25031209

- While revising your submission, please upload your figure files to the Preflight Analysis and Conversion Engine (PACE) digital diagnostic tool. PACE helps ensure that figures meet PLOS requirements.

Authors’ reply: We have appreciatively uploaded all our figure files to the PACE digital diagnostic tool, where they are accessible through the archive tab.

---

## [Decision Letter · Decision Letter 2]

6 Sep 2023

Interleukin-17, C-Reactive Protein, Neutrophil-to-Lymphocyte ratio, Lymphocyte-to-Monocyte ratio, and lipid profiles in healthy menopausal women with or without hot flashes: A cross-sectional study

PONE-D-22-24749R2

Dear Dr. Rezasoltani

We’re pleased to inform you that your manuscript has been judged scientifically suitable for publication and will be formally accepted for publication once it meets all outstanding technical requirements.

Kind regards,

Milad Khorasani, PhD

Academic Editor

PLOS ONE

Additional Editor Comments (optional):

Reviewers' comments:

Reviewer's Responses to Questions

**Comments to the Author**

1. If the authors have adequately addressed your comments raised in a previous round of review and you feel that this manuscript is now acceptable for publication, you may indicate that here to bypass the “Comments to the Author” section, enter your conflict of interest statement in the “Confidential to Editor” section, and submit your "Accept" recommendation.

Reviewer #2: (No Response)

Reviewer #4: All comments have been addressed

2. Is the manuscript technically sound, and do the data support the conclusions?

Reviewer #2: (No Response)

Reviewer #4: Yes

3. Has the statistical analysis been performed appropriately and rigorously? 

Reviewer #2: (No Response)

Reviewer #4: Yes

4. Have the authors made all data underlying the findings in their manuscript fully available?

Reviewer #2: (No Response)

Reviewer #4: Yes

5. Is the manuscript presented in an intelligible fashion and written in standard English?

Reviewer #2: (No Response)

Reviewer #4: Yes

6. Review Comments to the Author

Reviewer #2: The revision has been completed, and I am satisfied with the response. The author has addressed all of my previous comments.

Reviewer #4: The article is very well written and authors have satisfactorily addressed all the comments. Thank you and Congratulations

7. PLOS authors have the option to publish the peer review history of their article (what does this mean?). If published, this will include your full peer review and any attached files.

Reviewer #2: No

Reviewer #4: **Yes: **Bhuvanesh Kalal

---

## [Editor Report · Acceptance letter]

13 Nov 2023

PONE-D-22-24749R2 

Interleukin-17, C-Reactive Protein, Neutrophil-to-Lymphocyte ratio, Lymphocyte-to-Monocyte ratio, and lipid profiles in healthy menopausal women with or without hot flashes: A cross-sectional study   

Dear Dr. Rezasoltani:

I'm pleased to inform you that your manuscript has been deemed suitable for publication in PLOS ONE. Congratulations! Your manuscript is now with our production department. 

Kind regards, 

on behalf of

Dr. Milad Khorasani 

Academic Editor

PLOS ONE